# TLDR: Token-Level Detective Reward Model for Large Vision Language Models

**Deqing Fu**[1,2]    **Tong Xiao**[1]    **Rui Wang**[1]    **Wang Zhu**[1,2]
**Pengchuan Zhang**[1]    **Guan Pang**[1]    **Robin Jia**[2]    **Lawrence Chen**[1]
[1]Meta    [2]University of Southern California
deqingfu@usc.edu, lawrencechen@meta.com

## Abstract

Although reward models have been successful in improving multimodal large language models, the reward models themselves remain brutal and contain minimal information. Notably, existing reward models only mimic human annotations by assigning only one binary feedback to any text, no matter how long the text is. In the realm of multimodal language models, where models are required to process both images and texts, a naive reward model may learn implicit biases toward texts and become less grounded in images. In this paper, we propose a **T**oken-**L**evel **D**etective **R**eward Model (**TLDR**) to provide fine-grained annotations to each text token. We first introduce a perturbation-based method to generate synthetic hard negatives and their token-level labels to train TLDR models. Then we show the rich usefulness of TLDR models both in assisting off-the-shelf models to self-correct their generations, and in serving as a hallucination evaluation tool. We show that TLDR automatically trains a token-level likelihood optimization, and can improve the base model's performance significantly. Finally, we show that TLDR models can significantly speed up human annotation by 3 times to acquire a broader range of high-quality vision language data.

## 1 Introduction

Large vision language models (VLMs) are becoming increasingly powerful in generating human-like text, exemplified by models like GPT-4 family (OpenAI, 2024), Gemini and PaliGemma (Google, 2023; Beyer et al., 2024), LLaVA model family (Liu et al., 2024), and Llama 3 Vision models (Meta, 2024a). However, they are far from perfect and still suffer from generating hallucinated texts that are not grounded to the reference image. The need for accurate and interpretable reward models (RMs) to highlight the mistakes becomes increasingly critical. Traditional RMs, which are often binary classification models to provide one single score to evaluate entire outputs, have limitations in terms of interpretability and granularity. These models obscure the decision-making process of the model, making it challenging for humans to diagnose and improve performance at a fine-grained level.

To facilitate better interpretability and granularity, we propose a **T**oken-**L**evel **D**etective **R**eward (**TLDR**) model to offer a more interpretable alternative. By evaluating and assigning rewards at each token, rather than across entire sequences, TLDR enables greater transparency. This fine-grained approach allows for clearer identification of where a model excels in its output generation. Such interpretability is crucial not only for aligning model behavior with human expectations but also for improving human-AI interaction – a human annotator can swiftly fix the highlighted errors given by TLDR to make them correct because token-level evaluations allow for quicker identification of errors and more targeted improvements.

Additionally, a naive binary reward model could be biased towards text modalities – the longer the text, the higher the score, despite any internal hallucinations, making them less effective in multimodal contexts where visual information is essential. Our work aims to address this by constructing a reward model that is more visually grounded, incorporating multimodal cues to better evaluate model performance. The interpretability afforded by token-level granularity helps facilitate this grounding, ensuring that visual and textual signals are both considered in reward calculations. Ab-

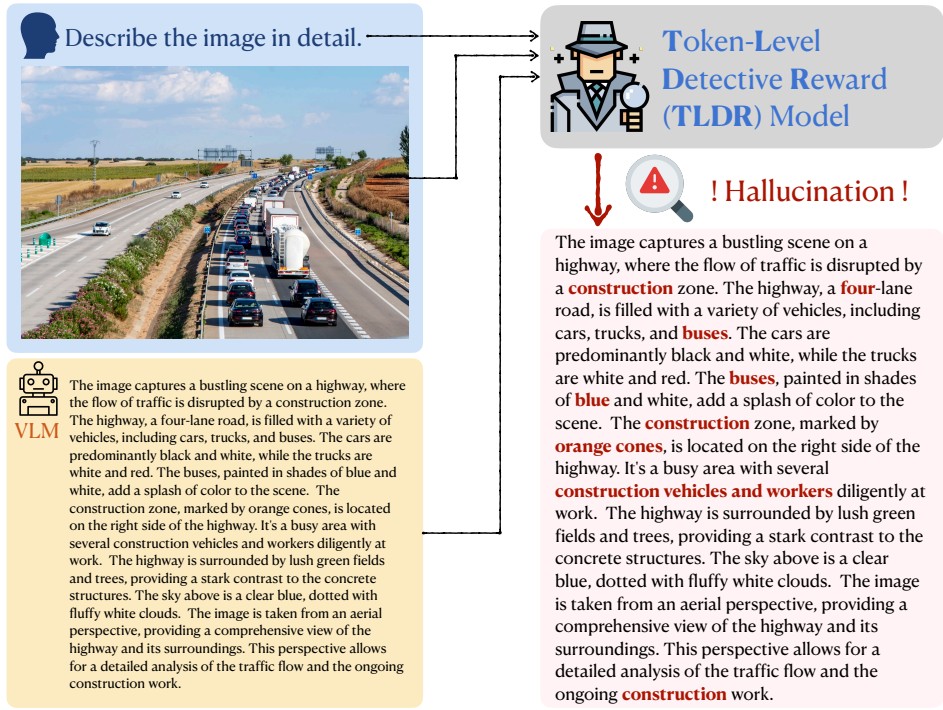

Figure 1: **T**oken-**L**evel **D**etective **R**eward (***TLDR***) Model. It can be used as hallucination detection, and to prompt models to self-correct with these detection. TLDR can also speed up human annotation speed to fix slightly mistaken image captions, to create high-quality vision language data.

lation studies in training TLDR models shown in Tables 10 and 11 verifies the claim by showing the sharp improvement of the RM's performance by further finetuning the linear projection of the VLM — projecting visual features given by the vision encoder to the textual embedding space. Unlike existing token-level RL methods that primarily serve as auxiliary post-training evaluation tools, TLDR is designed for on-policy RLHF training in vision-language models, bridging the gap between fine-grained supervision and model optimization.

Moreover, token-level reward models have the potential to enhance existing methods for model improvement. By providing detailed feedback on a token-by-token basis, these models enable more effective self-correction and refinement in generated outputs. A more granular understanding of errors can improve the performance of fine-tuning techniques such as DPO (Rafailov et al., 2024) and PPO (Schulman et al., 2017), where strong and interpretable reward signals are essential for optimizing model behavior. In Section 5.4, we show the TLDR model is automatically a likelihood training objective, that simultaneously improve the base vision language model behind the RM.

In a summary, TLDR model aims to develop a reward model that not only reflects human preferences more accurately but also enhances usability and interpretability. By improving the transparency of the reward mechanism at the token level, we provide a tool that facilitates faster feedback, better self-correction, simple likelihood finetuning, and a trustworthy hallucination evaluation metric. Our work establishes the first token-level reward model specifically tailored for vision-language models, enabling both hallucination detection and self-correction, while also naturally integrating into reinforcement learning frameworks. By unifying token-level reward modeling with vision-language tasks, TLDR significantly improves grounding capabilities and reduces hallucination rates.

## 2   RELATED WORK

**Reinforcement Learning from Human Feedback and Reward Model.**   Using Reinforcement Learning to align lan- guage models with human feedback or preferences (RLHF, Christiano et al., 2017; Ziegler et al., 2020) has led to phenomenal improved large language models such as Chat-GPT (Ouyang et al., 2022) and LLaMA 3 (Meta, 2024a). Similar RLHF techniques are used to

align large vision language models Liu et al. (2023) and text-to-image models (Lee et al., 2023; Sun et al., 2023a) as well. In general, RLHF involves training a **reward model** on user preference data collected from human annotators (Wang et al., 2024a). Given a reward model, a policy can be learned using Reinforcement learnings algorithms like Proximal Policy Optimization (PPO, Schulman et al., 2017). Alternatively, recent works have developed Direct Policy optimization (DPO, Rafailov et al., 2024) wherein reward models are mainly used for finding chosen and rejected pairs. DPO alignment for vision-language models is also well studied (Yu et al., 2023b; 2024; Li et al., 2024; Pi et al., 2024). Reward models themselves are also evolving including process reward models (Luo et al., 2023), step-wise reward models (Havrilla et al., 2024), etc. Recent works also attempt span-level or token-level detection but they are limited to the language domain (Yoon et al., 2024; Yang et al., 2024), and they are mostly sentence-level (Wu et al., 2023; Niu et al., 2024; Mishra et al., 2024) or need factual augmentations (Sun et al., 2023b). Unlike previous token-level reward works (Yoon et al., 2024) on for *offline RLHF*, we propose the first unified token-level reward model, TLDR, which establishes the stage for vision-language *on-policy RLHF training with token-level reward*. TLDR facilitates image-to-text hallucination detection (Rohrbach et al., 2018; Li et al., 2023; Jing et al., 2024; Lovenia et al., 2024) with efficient human correction, and improves downstream vision-language grounding performance.

**Synthetic Data and Hard Negative Mining.** Several NLP datasets have gathered instances of the negative class for their task. Many relied on human annotation, for example, unsupported claims in fact verification (Aly et al., 2021; Wadden et al., 2020), non-entailed hypotheses in NLI (Bowman et al., 2015), unanswerable questions (Rajpurkar et al., 2018). Some have used heuristics and external knowledge sources to automatically mine negative examples (Lee et al., 2021; Wright et al., 2022). Finally, there are hybrid approaches where candidate negative examples are first automatically generated and then manually verified (Wadden et al., 2022), or candidate negative examples are synthesized by model perturbation and verified by the same model (Fu et al., 2023).

**Large Vision Language Models and Evaluation.** There has been a plethora of recent developments in VLMs; they can be broadly categorized by their methods for representing visual modalities. A representative approach involves *tokenizing* visual inputs to be jointly trained with language inputs (Yu et al., 2023a; Google, 2023; Chameleon, 2024). Another line of work processes continuous visual features by directly projecting them to the language embedding space via a learnable function (Liu et al., 2024; Bavishi et al., 2023). At the core of these design choices is the hardness in representing visual features, which has been reported by several early studies (McKinzie et al., 2024) to be the key bottleneck towards better vision-language foundation models. Various benchmark datasets beyond MMMU (Yue et al., 2024) were proposed targeting these bottlenecks, such as BLINK (Fu et al., 2024b) and Vibe-Eval (Padlewski et al., 2024) for visual reasoning, IsoBench (Fu et al., 2024a) and MathVista (Lu et al., 2024) for algorithmic visual problem solving. At the essence of common mistakes made by VLMs, hallucination is a significant portion, and thus TLDR is designed as an evaluation tool to measure models' hallucination rate (see Table 2).

## 3 PROBLEM SETUP

A multimodal query-response instance $x = (m, p, d)$ is usually equipped with three elements, an image $m$, a user text prompt $p$, and a text response $d$. Training a reward model involves training a classifier $\rho(m, p, d) \in \{0, 1\}$ to predict human preference on the target response $d$ given the image $m$ and prompt $p$. In contrast, to have better granularity for the reward model, a token-level version is needed. Instead of training a point-wise scalar classifier, which only assigns a singular value to the instance $x$, it assigns values for every token of the target response $d = \{e_1, \cdots, e_N\}$ with $N = |d|$ tokens in total, where $|\cdot|$ denotes the number of tokens of any text sequence. It involves training the TLDR model $\mathbb{P}_\gamma$ to match fine-grained rewards. TLDR model's prediction can be written as

$$\gamma(m, p, d) = \Big(\gamma(e_1 \mid m, p, d), \cdots, \gamma(e_N \mid m, p, d)\Big) \in [0, 1]^N \tag{1}$$

where for any token $e$ in text response $d$, $\gamma(e \mid m, p, d) = \begin{cases} 1, & \text{if } \mathbb{P}_\gamma(e \mid m_k, p_k, d_k) > \theta \\ 0, & \text{otherwise} \end{cases}$, with

some threshold $\theta$ usually set to 0.5 if not otherwise mentioned.

Given the image, prompt, and response tuples $(m_k, p_k, d_k)$ in an evaluation set $\mathcal{S}$, we design three accuracy metrics on the TLDR model's performance.

**Token-Level Accuracy.** For each instance $x_k = (m_k, p_k, d_k)$, we are given true token-level labels $\gamma^\star(m_k, p_k, d_k) = \left( \gamma^\star(e_1), \cdots, \gamma^\star(e_N) \right)$. Then we define token-level accuracy as

$$\mathcal{A}_{\mathrm{T}}(\gamma, \mathcal{S}) = \frac{1}{|\mathcal{S}|} \sum_{(m_k, p_k, d_k) \in \mathcal{S}} \frac{1}{|d_k|} \sum_{e \in d_k} \mathbb{1} \left\{ \gamma^\star(e) = \gamma(e \mid m_k, p_k, d_k) \right\} \tag{2}$$

**Sentence-Level Accuracy.** Similar to the token-level accuracy but with each response $d$ broken into sentences $d = \{s_1, \cdots, s_{c(d)}\}$ with $s_j = \{e_{n_{j-1}+1}, \cdots, e_{n_j}\}$ where $n_j$ is the token position of the $j$-th period. We also let $n_0 = 0$ as the starting position. We also $c(\cdot)$ as the function of counting number of sentences in any text response $d$. Then we define sentence-level accuracy as

$$\mathcal{A}_{\mathrm{S}}(\gamma, \mathcal{S}) = \frac{1}{|\mathcal{S}|} \sum_{(m_k, p_k, d_k) \in \mathcal{S}} \frac{1}{c(d_k)} \sum_{j \in 1, \cdots, c(d_k)} \mathbb{1} \left\{ \prod_{e \in s_j} \gamma^\star(e) = \prod_{e \in s_j} \gamma(e \mid m_k, p_k, d_k) \right\} \tag{3}$$

A visual illustration on grouping tokens into sentences is shown in Figure 3.

**Response-Level Accuracy.** We define response level prediction as

$$\rho_\gamma(m, p, d) = \prod_{e \in d} \gamma(e \mid m, p, d) \quad \text{and} \quad \rho^\star(m, p, d) = \prod_{e \in d} \gamma^\star(e) \tag{4}$$

Then we compare with the ground truth labels $\rho^\star$ to define response-level accuracy as

$$\mathcal{A}_R(\rho_\gamma, \mathcal{S}) = \frac{1}{|\mathcal{S}|} \sum_{(m_k, p_k, d_k) \in \mathcal{S}} \mathbb{1} \left\{ \rho_\gamma(m_k, p_k, d_k) = \rho^\star(m_k, p_k, d_k) \right\} \tag{5}$$

Notably, the response-level accuracy also applies to response-level naive reward models $\rho$ under the same definition. We will further compare the response-level accuracy of TLDR model $\mathcal{A}_R(\rho_\gamma, \mathcal{S})$ and that of the naive model $\mathcal{A}_R(\rho, \mathcal{S})$ in Section 5.

Besides accuracy metrics, which could be biased when labels are imbalanced, especially in the token-level cases where a majority of the tokens are neutral tokens – words that won't affect response quality, we report **mean Average Precision** (mAP). Together with normal mAP metrics, for which we call mAP(pos) we also design a flipped version mAP(neg), where both the predicted labels and the ground-truth labels are flipped so that we pay more attention to tokens with negative labels. Because the token-level annotations are highly imbalanced, with more than 95% of them are positive tokens (see the example in Figure 1), the mAP(neg) is a more meaningful average precision metric. We report both mAP scores in the tuple format **mAP(neg|pos)**.

## 4 Synthetic Data Generation

Although aligning models toward user preference has become a standard post-training procedure, open-sourced user preference data, especially multimodal ones, are increasingly difficult to source. What is even worse, existing user preference data are mostly coarsely annotated as each instance is only given one label: thumb down or thumb up. To gather large amounts of fine-grained token-level preference data, we adopt the procedure by perturbing gold labels, inspired by Fu et al. (2023).

In this work, we mainly focus on two types of tasks: dense captioning and visual question answering (VQA). For VQA data, we synthesize hard negatives from **Visual Genome (VG100K)** dataset (Krishna et al., 2016), which contains 108,077 images with over 1.7 million question-answer pairs. For any VQA instance $x$ with image $m$, question $p$ and answer $d$, we prompt a pretrained large language model $\phi$, which takes the original question $p$ and answer $d$, and is instructed to generate a perturbed answer $d' = \phi(p, d)$ so that it's the *wrong* answer to the question $p$ given the image $m$, i.e., $\mathbb{P}(d' \mid m, p) = 0$. Notably, the model used for perturbation $\phi$ is text-only, without seeing the image $m$ to mitigate any visual biases. In this work, we use `Llama-3.1-70B` (Meta, 2024a) as the perturbation model $\phi$.

Admittedly, as dense captions are relatively longer than VQA samples, the amount of data available is more limited, and the hard negative synthesis process could be more versatile and more

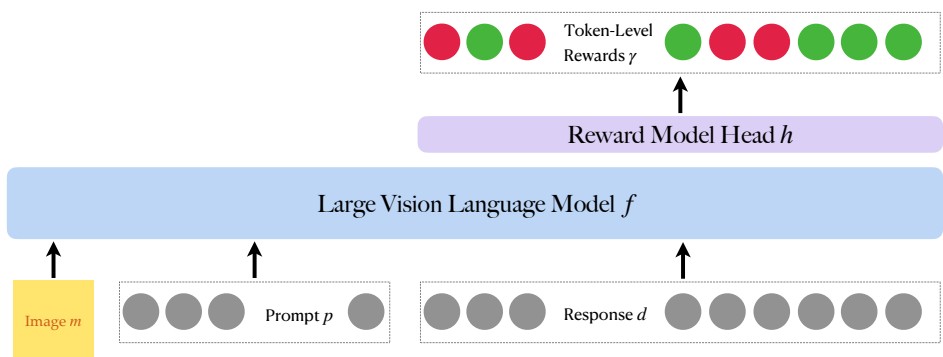

Figure 2: **TLDR Model Architecture.** For any instance with image $m$, prompt $p$, and a response $d$, they are passed altogether into the large VLM backbone $f$ without the language model head $\ell$. Then a shared reward model head $h$ is applied to every token $e_k$ of the response $d$ to have binary predictions $\gamma(e_k)$ to determine if $e_k$ is a good token or a bad token.

complicated. We synthesize hard negatives from **DOCCI** dataset (Onoe et al., 2024), which contains over 15,000 images and their corresponding dense captions. However, the amount of captions here is much fewer than the amount of VQA instances. To compensate the imbalance, we use `Llama-3.1-70B` to aggregate VQA instances to dense captions. For each image $m$ in the VG100K dataset, it's equipped with on average 16 question-answer pairs $\{(p_1, d_1), \cdots, (p_k, d_k)\}$. The text-only LLM is prompted to combine them into a dense caption $d$ for image $m$. Now, combining DOCCI and VG100K's synthetic caption, we have over 120,000 image-caption pairs. As prior work identifies (Lin et al., 2024), vision language models usually suffer in the following eight taxonomies. We enumerate each of them with an illustrated example pair.

    I. SPATIAL RELATIONSHIP: A is **left** to B $\longleftrightarrow$ A is **right** to B.
    II. VISUAL ATTRIBUTE: A is **yellow**. $\longleftrightarrow$ A is **blue**.
    III. ATTRIBUTE BINDING: A is **blue** and B is **yellow**. $\longleftrightarrow$ A is **yellow** and B is **blue**.
    IV. OBJECT IDENTIFICATION: A **dog** chasing a ball. $\longleftrightarrow$ A **cat** chasing a ball.
    V. COUNTING: **One** duck is swimming. $\longleftrightarrow$ **Four** ducks are swimming.
    VI. SMALL OBJECT: **Cirrostratus cloud** in the sky. $\longleftrightarrow$ **Clear** sky.
    VII. TEXT OCR: A shirt writes **heavy fog**. $\longleftrightarrow$ A shirt writes **happy frog**.
    VIII. COUNTERFACTUAL: A soldier. $\longleftrightarrow$ A soldier **has no sword in hand**.

For each taxonomy $t$, and for each instance with image $m$ and caption $d$, we use a prompt-engineered text-only LLM $\phi_t$ to generate a perturbed caption $d' = \phi_t(d)$ so that $d'$ is a minimal-edit from $d$. Furthermore, we prompt-engineer another LLM $\phi_c(d, d', t)$ to check they are not paraphrases and their difference lies in the desired taxonomy $t$. If they fail $\phi_c(d, d', t)$, we discard the perturbation. For instance, not every image has text written in it, so there is no way to generate perturbations focused on text OCR.

For either VQA or dense caption tasks, once we obtain successful perturbation $d' = \{e'_1, \cdots, e'_{|d'|}\}$, we compute the differences to the original text $d = \{e_1, \cdots, e_{|d|}\}$ to obtain the token-level label $\gamma^*(e'_k) \in \{0, 1\}$ depending on whether $e'_k$ appear in the neighborhood of $e_k$ in $d$ or not. Since the original text $d$ is human written, all of its tokens have positive label $\gamma^\star(e_k) = 1, \forall e_k \in d$.

We include prompts for perturbation in Appendices A.1 to A.3 and the statistics of our synthetic data in Table 7 at Appendix A.4.

## 5 EXPERIMENTS

### 5.1 TRAINING TLDR MODELS

**Model Architecture.** As shown in Figure 2, we use `PaliGemma-3B-Mix-448` (Beyer et al., 2024) and `Llama-3.2-11B-Vision` (Meta, 2024b) as our backbone pretrained large Vision

| Base Model | Reward Model | Token-Level Accuracy $\mathcal{A}_T$ | Sentence-Level Accuracy $\mathcal{A}_S$ | Response-Level Accuracy $\mathcal{A}_R$ | mAP (neg\|pos) |
|---|---|---|---|---|---|
| `PaliGemma-3B` | TLDR | 98.6 | 86.5 | **83.1** | (41.3\|99.8) |
| | Naive | — | — | 81.1 | — |
| `Llama-3.2-11B-Vision` | TLDR | 98.9 | 90.8 | **88.2** | (45.7\|99.8) |
| | Naive | — | — | 86.7 | — |
| `GPT-4o` | Prompting | 95.5 | 66.9 | 52.9 | (19.7\|98.1) |
| *Random Guess* | — | — | — | *50.0* | — |

Table 1: **Performance of the TLDR model.** As a reference, we include scores for the response-level naive reward model trained on the same dataset. Best response-level accuracy are highlighted in **bold** conditioning on the same base model. We find that the TLDR model has a slightly higher response-level accuracy when compared to the naive binary RM. A break down of response-level accuracy by taxonomy is shown in Table 10. Having a base model with larger number parameters also gives higher accuracies across all granularity.

**Language Model $f$.** Instead of using the pretrained language modeling head $\ell$ which maps the last hidden states to the vocabulary logits, we train a new *reward model head* $h : \mathbb{R}^{D_{\text{hidden}}} \to \mathbb{R}$ to map the last hidden states with dimension $D_{\text{hidden}}$ to a scalar logit for each token from the response $d$. For any instance equipped with image $m$, prompt $p$ and response $d$, we denote $|\cdot|$ as the number of tokens after tokenization for either image or text modality. The backbone language model gives the last hidden states $\mathbf{H} = f(m, p, d) \in \mathbb{R}^{(|m|+|p|+|d|) \times D_{\text{hidden}}}$ and the probability of being positive for $k$-th token $e_k$ in the response $d$ is given by

$$\mathbb{P}_\gamma(e_k \mid m, p, d) = \sigma\left(h\left(\mathbf{H}_{\star, (|m|+|p|+k)}\right)\right), \quad \text{where } \sigma \text{ is the Sigmoid function.} \quad (6)$$

In our setup, the reward model head $h$ is a simply linear layer with $(D_{\text{hidden}} + 1)$ parameters. We provide more detailed training setups and hyperparameters in Appendix B.

**Training.** As the PaliGemma report (Beyer et al., 2024) suggests, PaliGemma used full attention between input images and input texts, and only has autoregressive attention at generation. Similar to LLaVA model family (Liu et al., 2024), PaliGemma model $f$ has four major components: a 400M SigLIP (Zhai et al., 2023) vision encoder $f_{\text{enc}}$, a linear projection module $f_{\text{proj}}$ to align the vision features to the proper text embedding spacing, a Gemma-2B (Google, 2024) Transformer decoder $f_{\text{dec}}$, and a language model head $\ell$. We train our randomly initialized reward model head $h$, together with $f_{\text{proj}}$ and $f_{\text{dec}}$. For efficient training, we use LoRA (Hu et al., 2021) technique to update weights $\Theta_{\text{proj}}$ of $f_{\text{proj}}$ and $\Theta_{\text{dec}}$ of $f_{\text{dec}}$, so that $\Theta' = \Theta + \alpha_{\text{train}} AB$. We choose $\alpha_{\text{train}} = 128$ and $r := \text{rank}(A) = \text{rank}(B) = 512$ for all submodules $\Theta$. Models are trained with respect to the cross-entropy objective on every token of the response $d$, compared to the token-level label generated following Section 4.

In contrast, we compare with a naive reward model trained on the same training data and with the same hyperparameters. Although sharing the same architecture and parameter count, the naive reward model differs from the TLDR model as its cross-entroy loss is only computed at the last token of each response $d$, instead of on all tokens.

**Evaluation.** We evaluate TLDR model's performance on the synthetic data generated from the test split of the DOCCI dataset (Onoe et al., 2024). We measure the performance based on the metrics discussed in Section 3. As shown in Table 1, the TLDR model has slightly higher response-level accuracy than the naive binary RM. TLDR model has a 41.3 mAP(neg) and signals further room for improvements. A break-down of response-level taxonomy in Table 10 at Appendix B shows that, TLDR model performs the worst one spatial relationship taxonomy, and this resonances prior work that image grounding to spatial relationship is one of the hardest task for both image-to-text VLMs and text-to-image generations (Lin et al., 2024).

We conduct further human evaluation on token-level predictions on 100 samples from WinoGround (Thrush et al., 2022) images with captions generated by `MiniCPM`, `Phi-Vision-3.5` and `Qwen2-VL-7B`. With a special focus on false negative (FN) type of errors and averaged among three human annotators, we find the TLDR model has a sentence-level FN rate of 8.7%, 10.5% and 9.8%, respectively.

| | HALLUCINATION RATE (%) ↓ | | | | |
|---|---|---|---|---|---|
| MODELS | TOKEN-LEVEL | SENTENCE-LEVEL | RESPONSE-LEVEL | MMMU ↑ | MEGA-Bench ↑ |
| GPT-4o | **0.016** | 0.23 | 1.62 | **69.1** | **54.1** |
| Llama-3.2-90B-Vision | 0.017 | **0.19** | **1.23** | 60.3 | / |
| GPT-4o-mini | 0.030 | 0.38 | 2.12 | 59.4 | 43.0 |
| GPT-4-Turbo-Vision | 0.033 | 0.62 | 3.12 | 56.8 | / |
| Qwen2-VL-7B | 0.061 | 0.48 | 1.96 | 54.1 | 35.9 |
| Qwen2-VL-2B | 0.066 | 0.72 | 1.70 | 41.1 | 22.3 |
| MiniCPM-Llama-3-V2.5 | 0.067 | 0.81 | 3.62 | 45.8 | 22.8 |
| Llama-3.2-11B-Vision | 0.073 | 0.85 | 1.88 | 50.7 | 18.0 |
| Phi-Vision-3.5-Instruct | 0.261 | 2.65 | 9.25 | 43.0 | 25.3 |
| PaliGemma-3B-Mix-448 | 4.444 | 5.96 | 17.50 | 27.3 | / |

Table 2: **Hallucination Evaluation.** We prompt each model with image captioning instructions on 800 images from WinoGround (Thrush et al., 2022), and use TLDR to compute the hallucination rates (the lower the better), with respect to various levels of granularity. Model performances are sorted by token-level rate. We observe that GPT-4o is overall the best model with the least token-level hallucinations but Llama-3.2-90B-Vision has better sentence-level and response-level hallucinations. We also include self-reported MMMU (Yue et al., 2024) results to demonstrate their significant correlation with hallucination rates: the Pearson correlation between $-\log \mathcal{H}_T$ and MMMU score is 0.902 with a $p$-value of $3.45 \times 10^{-4}$. The correlation can also be visually observed in Figure 6.

Figure 3: **Level of Granularity in Hallucination Rate.** Using the example from Figure 1, we can easily compute token-level hallucination rates following Eq. 7. Then tokens are grouped into sentences which are separated by period marks. An entire sentence with at least one bad token is highlighted as a bad sentence. Then the sentence-level hallucination rate of one response is calculated by counting the proportion of bad sentences. Similarly, if there is at least one bad token in the response, the entire response is a bad one. Hallucination rates are averaged over an entire evaluation set to determine the overall hallucination rates of a model.

**Ablation Study.** As discussed earlier, we finetune a randomly initialized reward model head $h$, together with LoRA efficiently finetuning the multimodal projection layer $f_{\text{proj}}$ and the Transformer decoder $f_{\text{dec}}$. In the section, we ablate the necessity of LoRA finetuning $f_{\text{proj}}$ and $f_{\text{dec}}$.

As shown in Table 11 at Appendix Appendix B.2, both linear projection module and the decoder module are worth training and they work together to facilitate the performance of our TLDR model. One interesting observation is training $f_{\text{proj}}$ on top of $f_{\text{dec}}$ barely improves token-level accuracy or mAP(pos). But it helps with both sentence-level and response-level accuracy, and increases mAP(neg) significantly. This could be explained by that finetuning $f_{\text{proj}}$ reduces the model's false negative rates on tokens because it's more visually grounded by tuning the projection from visual space to textual space.

## 5.2 HALLUCINATION EVALUATION WITH TLDR MODELS

Since our TLDR model provides token-level predictions, we can use it to compute a model's hallucination rates without requiring ground truth labels. Given image $m$, prompt $p$, and a model for evaluation $\xi$. We first obtain model's response $\hat{d} = \xi(m, p)$ with tokens $\{\hat{e}_1, \cdots, \hat{e}_{|\hat{d}|}\}$. Then our TLDR model gives its prediction $\gamma(m, p, \hat{d}) = \left( \gamma(\hat{e}_1 \mid m, p, \hat{d}), \cdots, \gamma(\hat{e}_{|\hat{d}|} \mid m, p, \hat{d}) \right)$. Then the

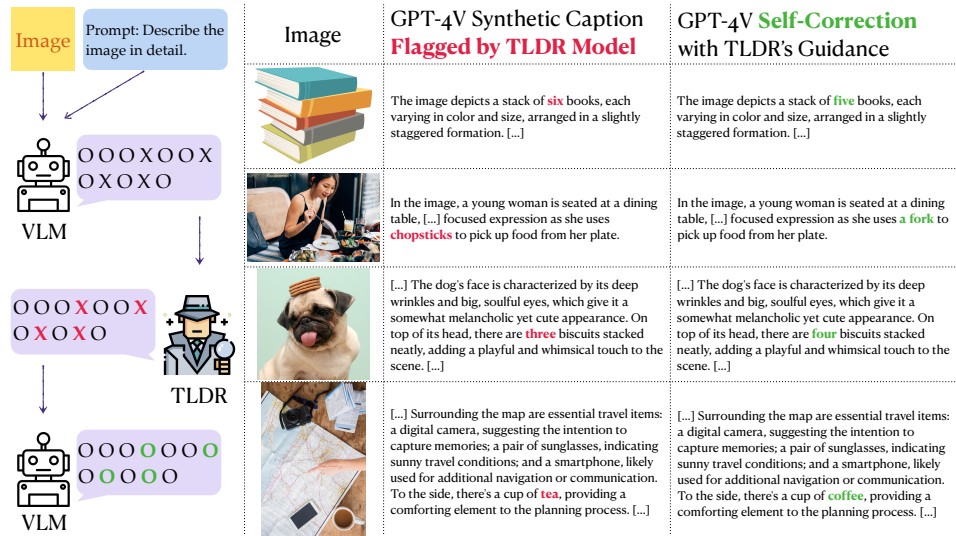

Figure 4: TLDR model can guidance existing VLMs to **Self-Correct** their hallucination when generating captions for images from WinoGround (Thrush et al., 2022).

token-level hallucination rate for this instance is $\frac{1}{|\hat{d}|} \sum_{\hat{e}_k \in \hat{d}} \gamma(\hat{e}_k \mid m, p, \hat{d})$. Similar to sentence-level and response-level accuracy defined in Section 3, we can have sentence-level and response-level hallucination rates as well. Their definitions are shown in Figure 3.

Now we can have the token-level hallucination rates for the model $\xi$ given any dataset $\mathcal{S}$ as follows

$$\mathcal{H}_T(\xi, \mathcal{S}) = \frac{1}{|\mathcal{S}|} \sum_{(m,p) \in \mathcal{S}} \frac{1}{|\underbrace{\xi(m,p)}_{\hat{d}}|} \sum_{\hat{e}_k \in \hat{d}} \gamma(\hat{e}_k \mid m, p, \hat{d}) \tag{7}$$

We evaluate Llama-3.2-Vision (Meta, 2024b) with 11B and 90B versions, GPT-4o, 4o-min and GPT-4 turbo vision (OpenAI, 2024), MiniCPM (Yao et al., 2024), PaliGemma (Beyer et al., 2024), Qwen2-VL (Wang et al., 2024b) with 2B and 7B versions, and Phi 3.5 Vision (Abdin et al., 2024) with our TLDR Model.

As shown in Table 2, GPT-4o is overall the best model with the least amount of hallucinations among all granularity. We also observe a strong correlation between model's hallucinate rates and its visual understanding and reasoning performance evaluated by MMMU. We conjecture for any VLM $\xi$,

$$\text{Performance}(\xi) \propto -\log \mathcal{H}_T(\xi) \tag{8}$$

## 5.3 SELF-CORRECTION WITH TLDR MODELS

Hallucination detection and evaluation (Zhou et al., 2024; Jing & Du, 2024; Yin et al., 2023) is a small leap forward. The most exciting usage of TLDR model is to guide models with **self-correction** and to be more grounded to the images by taking another guided look. Token-level annotations from TLDR model can enhance a vision-language model's ability to self-correct by providing detailed, granular feedback on specific parts of its output. These annotations allow the model to break down its response into smaller, interpretable units, aligning each token with visual and textual cues. By analyzing where errors occur at the token level—whether in object recognition, attribute descriptions, or language syntax—the model can more precisely identify the source of the mistake. Additionally, token-level feedback can guide the model to better align its language generation with the visual context, improving coherence and factual accuracy in its self-correction process. For self-correction experiments, we chose `GPT-4V` because it provides a strong baseline for assessing reward-driven refinements, while also including `Llama-3.2-90B-Vision` and `Qwen2-VL-7B` to showcase TLDR's effectiveness across diverse architectures.

In this section, we evaluate on **WinoGround** (Thrush et al., 2022) dataset to show whether given extra token-level annotation cues, the vision language model is able to self-correct its own halluci-

| MODEL | Guidance Given By | # Samples | # Samples Flagged by RM | # Self-Corrected | Win | Tie | Loss |
|---|---|---|---|---|---|---|---|
| GPT-4V | TLDR (ours) | 800 | 25 | 21 | 12 | 7 | 2 |
|  | Naive |  |  | 15 | 2 | 11 | 2 |
| Llama-3.2-90B | TLDR (ours) | 800 | 10 | 8 | 6 | 1 | 1 |
|  | Naive |  |  | 6 | 3 | 1 | 2 |
| Qwen2-VL-7B | TLDR (ours) | 800 | 25 | 16 | 9 | 5 | 2 |
|  | Naive |  |  | 9 | 3 | 5 | 1 |

Table 3: **Self-Correction with the Guidance of TLDR Model.** A target model (`GPT-4V`, `Llama-3.2-90B` or `Qwen2-VL-7B`) is used to generate captions for 800 images from WinoGround, some of the captions are flagged by the reward model as containing bad tokens. When prompted to self-correct, extra guidance from TLDR helps the target model correct more of its own hallucinations with larger win rates.

| MODELS | TASKS | | | | |
|---|---|---|---|---|---|
| | BLINK ↑ | | | IsoBench ↑ | |
| | Count | Spatial Relation | Object Localize | Function Parity | Chess Winner |
| PaliGemma-3B | 69.2 | 78.3 | 45.9 | 41.4 | 45.1 |
| + TLDR ($\tau = 0.25$) | **71.7** | 80.4 | **47.5** | **45.1** | 45.1 |
| + TLDR ($\tau = 0.5$) | **71.7** | **81.1** | 42.6 | 44.3 | **47.5** |
| + TLDR ($\tau = 1$) | 12.5 | 2.1 | 42.6 | 34.4 | 44.8 |
| Llama-3.2-11B-Vision | 55.0 | 61.5 | 60.7 | 34.9 | 45.5 |
| + TLDR ($\tau = 0.25$) | **67.5** | 65.0 | **67.2** | **35.4** | 43.6 |
| + TLDR ($\tau = 0.5$) | 65.8 | **65.7** | 59.0 | 33.3 | **47.9** |
| + TLDR ($\tau = 1$) | 61.7 | **65.7** | 56.6 | 35.1 | 39.4 |

Table 4: **Training TLDR model automatically gives a better vision language model for VQA.** We evaluate 3 versions of TLDR backbone model with different scales of LoRA $\alpha$. They are distinguished by $\tau = \alpha_{\text{infer}}/\alpha_{\text{train}}$, the proportion of $\alpha$ at inference and training time. We find that when $\tau = 0.25$, it could improve the PaliGemma model's performance by at most 3.7 points and could improve Llama 3.2 model's performace by at most 12.5 points.

| MODELS | HALLUCINATION RATE (%) ↓ | | |
|---|---|---|---|
| | TOKEN-LEVEL | SENTENCE-LEVEL | RESPONSE-LEVEL |
| PaliGemma-3B | 4.444 | 5.96 | 17.50 |
| + TLDR ($\tau = 0.10$) | 0.991 | 3.80 | 10.53 |
| + TLDR ($\tau = 0.25$) | **0.172** | **1.13** | **3.96** |
| Llama-3.2-11B-Vision | 0.073 | 0.85 | 1.88 |
| + TLDR ($\tau = 0.10$) | 0.078 | **0.69** | 2.71 |
| + TLDR ($\tau = 0.25$) | **0.066** | 0.74 | **1.72** |

Table 5: **Training TLDR model automatically gives a better vision language model with *less hallucination rate*.** Similar to Table 4 and the setup in Table 2, we evaluate TLDR backbone models' hallucination rate on the dense image captioning task. Models are able to reduce the hallucination rates, especially for less-performing models such as PaliGemma.

nations. Out of 800 captions generated by `GPT-4V` for images in WinGround, TLDR model flags 25 of them as including hallucinated tokens. As shown in Table 3, when prompted with TLDR guidance, `GPT-4V` attempts to self-correct 21 out of 25, and when evaluated by human annotators, 12 of them are improved, 7 of them are tied, and 2 of them are worsened. On the contrary, when prompted to self-correct *without* TLDR's guidance, `GPT-4V` attempts to self-correct 15 out of 25 with only 2 wins, 11 ties, and 2 losses. Examples of `GPT-4V`'s self-correction results are shown in Figure 4. We include both prompt templates for self-correction, with and without TLDR's guidance in Appendix C.

## 5.4 TLDR AUTOMATICALLY TRAINS TOKEN-LEVEL LIKELIHOOD OPTIMIZATION

The purpose of building reward models is to improve the backbone large vision language model. We find that, a free by-product of training the TLDR model is that the backbone model's weights are simultaneously updated together with the reward model head. As discussed in Section 5, the linear projection $f_{\text{proj}}$ and the transformer decoder Gemma-2B $f_{\text{dec}}$ are both updated with LoRA weights during training TLDR. Now we attach back the original pretrained language model head $\ell$ to the backbone of the updated PaliGemma model with TLDR (by discarding the reward model head $h$), we obtain an updated vision language model. Now we evaluate whether this new model has improved from the orginal model, by evaluating on both in-distribution tasks from BLINK (Fu et al., 2024b) and out-of-distribution tasks from IsoBench (Fu et al., 2024a). At inference time, we adopt a different LoRA alpha to merge the weights, for weight $\Theta$ for any updated module, $\Theta' = \Theta + \alpha_{\text{infer}} AB$, where $A, B$ are trained LoRA weights. We find the proportion between $\alpha_{\text{infer}}$ and $\alpha_{\text{train}}$ could affect model performance significantly. We denote this proportion $\tau := \alpha_{\text{infer}}/\alpha_{\text{train}} \in [0, 1]$. With $\tau = 0$, we are evaluating the original model before training TLDR, and with $\tau = 1$, we are evaluating the model trained to provide token-level rewards. As shown in Table 4, a $\tau = 0.25$ gives the best performance.

Such automatic improvement is in fact that TLDR simultaneously trains the backbone VLM with likelihood optimization. The binary cross entropy objective on $\mathbb{P}(e \mid m, p, d)$ for any token $e$ simply promotes the model to generate $e$ if $\gamma^\star(e) = 1$, i.e., $e$ is a good token; and suppresses the model to generate it if $\gamma^\star(e) = 0$, i.e., $e$ is a bad token. As the linear projection layer $f_{\text{proj}}$ is also finetuned, the model is promoted to be more visually grounded, with an improvement for spatial relationship and chess winner identification, both of which requires complex spatial reasoning on images.

## 5.5 SPEEDING UP HUMAN ANNOTATION WITH TLDR MODELS

Human annotations, especially on dense captions, are costly and model generated captions are not trustworthy. Recent work such as PixelProse (Singla et al., 2024) has started releasing model generated dense captions with an ambition to use these captions to train better models. However, on a random sampling of 3,000 images from PixelProse, TLDR model detects 22.39% of the captions have hallucinated tokens, with a token-level hallucination rate of 0.83% and a sentence-level hallucination rate of 5.23%.

Nonetheless, it's always easier and cheaper for human annotators to **correct** an existing caption than writing long captions **from scratch**. Instead of using model's self-correction as designed in Section 5.3, human correction could be more rigorous to provide better captions. Human annotators are given the similar instruction as model self-correction prompts. Instead of comparing their caption correction quality – corrected captions are later cross checked by annotators to ensure quality – annotators are asked to time their annotation speed. Each annotator is assigned two set of distinct samples, one with TLDR guidance and the other without, and is asked to fix the caption and time themselves. As shown in Table 6, all three annotators share a similar annotation speed when if given TLDR guidance or no extra guidance. Most importantly, all three annotators have a 3 times speed up, which could in the future help with creating large bulk of vision language data with lower annotation costs.

| ANNOTATOR ID | AVERAGE ANNOTATION SPEED (SECONDS) WITH GUIDANCE OF | |
|---|---|---|
| | Binary Reward Model | TLDR Model |
| Annotator A | 101.7 | 31.2 |
| Annotator B | 109.1 | 32.9 |
| Annotator C | 121.3 | 34.4 |
| AVERAGE | 110.7 | 32.8 |

Table 6: **TLDR Speeds up Human Annotation by *3 Times* to Fix Synthetic Image Caption in PixelProse (Singla et al., 2024)**.

## 6 DISCUSSION AND CONCLUSION

In this paper, we introduced a **T**oken-**L**evel **D**etective **R**eward Model (**TLDR**) to provide fine-grained annotations for large vision language models. It is more interpretable than traditional naive binary reward models as TLDR can inform the user not only the response could be wrong but also *where* the response is wrong. Such feature enables many meaning usages, such as model's self-correction with TLDR guidance, and hallucination evaluation with TLDR annotations. We also presented a naive baseline where TLDR model is automatically a likelihood optimization method for its backbone vision language model, and the TLDR-tuned VLM is able to improve in several benchmarks. We believe a strong RM is a crucial basis for token-level DPO and PPO post-training. Finally, we show that, beyond guiding model's to self-correct, TLDR can also assist human annotators to fix synthetically generated image captions by improving the annotation speed by 3 times. Although our evaluation primarily focuses on hallucination mitigation, TLDR's token-level reward framework could be extended to broader safety and alignment tasks in vision-language generation.

Avenues ahead, we aim to design better human annotation interfaces under the human-computer interaction realm, to further reduce annotation overhead, to acquire large amount of high quality image captioning data with both positive human-corrected caption and negative model-generated caption. It will further facilitate us in designing token-level policy optimization methods. We believe our approach has the potential to advance the field of reward modeling and automatic evaluation. We hope that TLDR will make data annotation easier, and guide multimodal LLMs to hallucinate less. We hope our work can inspire further research to rethink the roles of reward models, and how their transparency, interpretability, and granularity can help advance the field of building better multimodal foundation models.

ACKNOWLEDGEMENT

DF would like to thank Richard Yuanzhe Pang for the initial discussion on token-level predictions and automated model critiques for large language models. The detective and robot figures used in Figures 1 and 4 are from `flaticon.com`.

**Getty Acknowledgement.** Images in the paper that originated from the WinoGround dataset (Thrush et al., 2022) are a compilation of assets, including ©Getty Images/Natasha Breen, Maki Nakamura, Jessica Peterson, Kundanlall Sharma, lacaosa, Alberto Bogo, Vu Le, Toson Rueangsuksut, Nisian Hughes, Tanja Walter, Douglas Sacha, PBNJ Productions, Glow Images, 10'000 Hours, zoranm, Marlene Ford, Westend61.

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

APPENDIX

## A  SYNTHETIC DATA GENERATION

### A.1  PROMPT FOR SYNTHESIZING IMAGE CAPTION FROM VQA

> **SYNTHESIZE CAPTION FROM VISUAL QUESTION ANSWERING**
>
> ```
> Your task is to convert a list of question-answer pairs to a
> descriptive paragraph.
> Keep in mind these rules:
> Do not start with greetings or salutations.  Simply return
> the new caption.
> Do not write anything else at the end of your response.
> Your crafted descriptive caption should be very faithful to
> the given question-answer pairs.
> Do not add any additional information that is not in the
> question-answer pairs.
> The parapgraph should not start with the photo, the image,
> the picture etc.
> Instead of saying, for example, the photo shows a cute koala
> bear sleeping on the tree, you should just say a cute koala
> bear is sleeping on the tree.
> For example, you are given the following question-answer
> pairs:
> Question:  What is the color of the mug?  Answer:  red.
> Question:  Where is the mug at?  Answer:  on the table.
> Question:  Is there anything written on the mug?  Answer:
> Yes.
> Question:  What is written on the mug?  Answer:  Hello World.
> Question:  What is the color of the texts?  Answer:  yellow.
> Your response should be a descriptive paragraph that
> aggregate the information from the question-answer pairs:
> Paragraph:
> A red mug sitting on the table has yellow texts written on
> it:  Hello World.
> ```

```
Now you are given the following question-answer pairs and are
asked to generate a paragraph:
Question:  ⟨Question⟩ Answer:  ⟨Answer⟩
Question:  ⟨Question⟩ Answer:  ⟨Answer⟩
  ⋮
Paragraph:
```

## A.2   PROMPTS FOR SYNTHESIZING VQA NEGATIVES

### PROMPT TEMPLATE FOR GENERATING VQA NEGATIVES

```
You are provided with a visual question-answer pair.  Your
task is to generate a wrong answer that is subtly different
from the original answer.
Keep in mind the following rules:
Keep formatting the same, such as sentence structure,
paragraph blocks, or newlines.
The changes should be subtle but concrete, replacing words
and phrases with opposite meanings or alternative options.
Do not start with greetings or salutations.  Simply return
the new answer.
Do not write anything else at the end of your response.
Do not fix any typos or grammar errors.  If there are any,
please ignore them.
The new answer should be nearly identical, other than 1 or 2
very small changes.
The changes should be very visually different.  The new
answer should be realistic.
More importantly, keep the wrong answer within the same
taxonomy as the original answer.
Here are some examples.
Question:  What color is the apple?  Answer:  red.  Wrong
answer:  green.
Question:  What is the spatial relationship between the man
and the chair?  Answer:  The man is sitting on a brown chair.
Wrong answer:  The man is sitting next to a brown chair.
Question:  How many apples are there?  Answer:  There are 5
apples.  Wrong answer:  There are 6 apples.
Now write your new answer:
Question:  ⟨Question⟩ Answer:  ⟨Answer⟩ Wrong answer:
```

### CHECK IF VQA NEGATIVE IS VALID

```
You are provided with a paragraph, and a question-answer
pair.  Your task is to determine if the answer is a valid
answer to the question given the paragraph.
The answer should be a simple yes or no.  Do not write
anything else at the end of your response.  Here are some
examples.
Paragraph:  The man is sitting on a brown chair.  Question:
What is the spatial relationship between the man and the
chair? Answer:  The man is sitting on a brown chair.  Valid
answer:  yes.
```

```
Paragraph:  There are six apples on the table and they are
all red.  One of the apples is rotten and is at the left side
of the table.  Question:  How many apples are there?  Answer:
There are 5 apples.  Valid answer:  no.
Paragraph:  On a sunny day, a man sailing a boat on the ocean
sees a fish jumping out of the water.  Question:  What is the
man doing?  Answer:  The man is fishing.  Valid answer:  no.
Now check this paragraph and the question answer pair:
Paragraph:  ⟨Paragraph⟩ Question:  ⟨Question⟩ Answer:  ⟨Answer⟩
Valid answer:
```

**Note**: we discard the synthetic negatives that are marked as *valid* because the perturbation is unsuccessful.

### A.3  PROMPTS FOR SYNTHESIZING IMAGE CAPTION NEGATIVES

We first present the general prompt for perturbation and then break these down into taxonomy-specific rules and in-context examples on all 8 taxonomies described in Section 4.

GENERAL PROMPT

```
You are provided with a caption to an image.  Your task is
to generate a new caption that is subtly different from the
original caption.
Keep in mind the following rules:
Keep formatting the same, such as sentence structure,
paragraph blocks, or newlines.
The changes should be subtle but concrete, replacing words
and phrases with opposite meanings or alternative options.
Do not start with greetings or salutations.  Simply return
the new caption.
Do not write anything else at the end of your response.
Do not fix any typos or grammar errors.  If there are any,
please ignore them.
The new caption should be nearly identical, other than 1 or 2
very small changes.
The changes should be very visually different.
The new caption should be realistic.
Most importantly, make the change with the following
taxonomy:  ⟨Taxonomy⟩
Here are the additional rules for the taxonomy
⟨Taxonomy-Specific Rules and In-Context Examples⟩
Here is the original caption:  ⟨Caption⟩
Now write your new caption:
```

SPATIAL RELATIONSHIP RULES AND IN-CONTEXT EXAMPLES

```
Change the spatial relationship of two objects in the given
caption.
The new spatial relationship should be different from the
original spatial relationship.
The new spatial relationship should be realistic, and is
visually different from the original spatial relationship.
You should fix the consistency of the caption.  If you change
the spatial relationship, you should also change the noun,
```

```
verb, etc.  so that there is no grammar error.
Do not change anything not related to spatial relationship,
even there are spatial relationship present.
Do not change other attributes of objects, such as color,
texture, material, etc.
If there are no spatial relationship present in the caption,
you should simply copy the original caption.

For example,
Original caption:  A man is sitting on the left to a coffee
table.
New caption:  A man is sitting on the right to a coffee
table.

Original caption:  A duck is swimming in a pool and a fish is
swimming underneath.
New caption:  A duck is swimming in a pool and a fish is
swimming on top of it.

Original caption:  A man is sitting in front of a car.
New caption:  A man is sitting in a car.

Original caption:  An apple is placed on an open book.
New caption:  An apple is placed under an open book.

Original caption:  There is a black cat.
New caption:  There is a black cat.
```

### VISUAL ATTRIBUTE RULES AND IN-CONTEXT EXAMPLES

```
Change the visual attributes of objects in the given caption.
The new visual attributes should be different from the
original visual attributes.
The new visual attributes should be realistic, and is
visually different from the original visual attributes.
You should fix the consistency of the caption.  If you change
the visual attributes, you should also change the noun, verb,
etc.  so that there is no grammar error.
Do not change anything not related to visual attributes, even
there are visual attributes present.

For example,
Original caption:  A man is sitting on a marble bench.
New caption:  A man is sitting on a wooden bench.

Original caption:  A red apple is placed on a table.
New caption:  A green apple is placed on a table.

Original caption:  A corgi dog is sitting on a chair.
New caption:  A corgi dog is sitting on a couch.
```

### ATTRIBUTE BINDING RULES AND IN-CONTEXT EXAMPLES

```
Change the attribute bindings of many objects in the given
caption.
```

Definition of attribute binding is that the attribute of
an object is bound to the object.  Changing the attribute
binding means swap the attributes of many different objects.
The new attribute bindings should be different from the
original attribute bindings.
The new attribute bindings should be realistic, and is
visually different from the original attribute bindings.
You should fix the consistency of the caption.  If you change
the attribute bindings, you should also change the noun,
verb, etc.  so that there is no grammar error.
Do not change anything not related to attribute bindings,
even there are attribute bindings present.
Do not add or deletes any objects and attributes other than
changing the attribute bindings.

For example,
Original caption:  A man is sitting on a bench and a woman is
sitting on a chair.
New caption:  A man is sitting on a chair and a woman is
sitting on a bench.

Original caption:  A red apple and a stack of blue books are
on a table.
New caption:  A blue apple and a stack of red books are on a
table.

Original caption:  An apple made of aluminum and a chair made
of wood are on display at the art museum.
New caption:  An apple made of wood and a chair made of
aluminum are on display at the art museum.

Original caption:  Two yellow cats are chasing one flurry
blue ball of yarn.
New caption:  Two blue cats are chasing one flurry red ball
of yarn.

### OBJECT IDENTIFICATION RULES AND IN-CONTEXT EXAMPLES

Change the object identifications in the given caption.
Definition for object identification is that the entity of an
object, e.g., a book, a man, a dog, a table, etc.
The new object should be different from the original object.
The new object should be realistic, and is visually different
from the original object.
If possible, you can make the new object subtly different
from the original object.  For example, change a corgi dog to
a dachshund dog.
You should fix the consistency of the caption.  If you change
the object, you should also change the noun, verb, etc.  so
that there is no grammar error.
Do not change anything not related to object identification,
even there are object identifications present.
Do not add or deletes any objects and attributes other than
changing the object identification.

```
Here are some valid examples
Original caption:  A man is sitting on a bench.
New caption:  A man is sitting on a chair.

Original caption:  A red apple is placed on a table.
New caption:  A red apple is placed on a bench.

Original caption:  A corgi dog is sitting on a chair.
New caption:  A dachshund dog is sitting on a couch.
```

### COUNTING RULES AND IN-CONTEXT EXAMPLES

```
Change the counting of one object in the given caption to a
different number.
The new number should be different from the original number.
The new number should be realistic, and is not so different
from the original number.
You should fix the consistency of the caption.  If you change
the number, you should also change the noun, verb, etc.  so
that there is no grammar error.
Only change the counting of one object in the caption.  Do
not change the number of other objects.
Do not change anything not related to counting, even there
are numbers present.
Do not change things related to written texts in quotation
marks.  For example, the original caption has A man wears a
shirt with text 'cute cat' written on it.  DO NOT change the
caption to A man wears a shirt with text 'cute cats' written
on it.
Do not change things ralated to time in the caption.  For
example, the original caption has The clock reads 13:00.  DO
NOT change the caption to The clock reads 14:00.
Do not change things related to proportions.  For example,
the orginal caption has The book covers 2/3 of the table.  Do
NOT change the caption to The book covers 3/4 of the table.
You can change the caption to Two books cover 2/3 of the
table instead.
If there are no counting in the caption, you should simply
copy the original caption.

Here are some valid examples
Original caption:  There are five cats on the table and they
are black.
New caption:  There are seven cats on the table and they are
black.

Original caption:  There are two dogs standing on the chairs,
one is white and one is black.
New caption:  There are three dogs standing on the chairs,
one is white and the other two are brown.

Original caption:  A side view of a Rouen duck that is brown
and tan and in some water.  It is facing to the right.
New caption:  A side view of two Rouen ducks that are brown
and tan and in some water.  They are facing to the right.
```

```
Original caption:  A pair of stop sign poles on a street.
New caption:  Three stop sign poles on a street.

Original caption:  The sky is blue.
New caption:  The sky is blue.
```

### SMALL OBJECTS RULES AND IN-CONTEXT EXAMPLES

```
Change the small and background objects in the given caption.
The new small and background objects should be subtly
different.  You can change their counts, size, shape, color,
etc.
You should ONLY change very small, neglible, background
objects, that are explictly described as so in the caption.
For example, you can pay attention to words like tiny, small,
mini, micro, nano, pico, femto, nano, micro, milli, etc.
If there are no small objects in the caption, you should
simply copy the original caption.

For example,
Original caption:  A man is sitting on a bench, in a library
with a white background board.  On the bookshelf, there is a
tiny crystal superman figure standing on a stack of books.
New caption:  A man is sitting on a bench, in a library with
a black background board.  On the bookshelf, there is a tiny
plastic batman figure standing in front of a stack of books.

Original caption:  A man is sitting on a bench, in a library
with a white background board.
New caption:  A man is sitting on a bench, in a library with
a black background board.
```

### TEXT OCR RULES AND IN-CONTEXT EXAMPLES

```
Change the text OCR in the image caption to a different text.
The new text should be different from the original text.
The new text should be realistic to the context, and is
visually different from the original text.
You should fix the consistency of the caption.  If you change
the text, you should also change the noun, verb, etc.  so
that there is no grammar error.
Do not change anything not related to text, even there are
texts present.
If there are no texts OCR present in the caption, you should
find a place to put some reasonable text OCR. If you can't
find a place to put some reasonable text OCR, you should
simply copy the original caption.

For example,
Original caption:  A man is wearing a shirt with texts
'SUPERMAN'.
New caption:  A man is wearing a shirt with texts 'BATMAN'.
```

```
Original caption:  The digital clock reads 12:00 AM.
New caption:  The digital clock reads 12:08 PM.

Original caption:  The road sign says 'STOP'.
New caption:  The road sign says 'YIELD'.

Original caption:  The board says 'Best College in the US'.
New caption:  The board says 'Best College in the World'.

Original caption:  A man wearing yellow shirt is sitting on
the bench.
New caption:  A man wearing yellow shirt with words 'Hello
World' is sitting on the bench.
```

COUNTERFACTUAL RULES AND IN-CONTEXT EXAMPLES

```
Change the caption with counterfactuals.
The new caption should be different from the original
caption.
The new caption should be realistic, and is visually
different from the original caption.  You should fix the
consistency of the caption.  If you change the caption, you
should also change the noun, verb, etc.  so that there is no
grammar error.
Do not change anything not related to counterfactuals, even
there are counterfactuals present.
Do not add or deletes any objects and attributes other than
changing the counterfactuals.
If it's hard to put in counterfactual, you should simply copy
the original caption.

For example,
Original caption:  A man is sitting on a bench.
New caption:  A man is not sitting on a bench.

Original caption:  A red apple is placed on a table.
New caption:  A red apple is not placed on a table.

Original caption:  A soldier.
New caption:  A soldier has no sword in hand.
```

## A.4   STATISTICS OF DATA

| TASK | Data Source | TAXONOMY | # POSITIVE | # NEGATIVE | TRAIN SET PROPORTION (%) |
|---|---|---|---|---|---|
| VQA | VG100K | — | 1,179,007 | 1,179,007 | |
| Image Caption | Synthetic Caption from VG100K | Spatial Relation
Visual Attribute
Attribute Binding
Object Identification
Counting
Small Object
Text OCR
Counterfactual | 94,684 | 45,225
86,366
59,219
75,328
75,156
80,455
84,164
57,153 | 80% |
| Image Caption | DOCCI | Spatial Relation
Visual Attribute
Attribute Binding
Object Identification
Counting
Small Object
Text OCR
Counterfactual | 14,639 | 8,867
13,811
13,561
10,618
10,491
11,680
13,366
12,844 | 65% |

Table 7: **Statistics of Data.** Overall, we have over 1M VQA data with both positive and negative answers, and over 100K caption datapoints with 650K negative captions. We oberserve that we have the least amount of spatial relationship data, because spatial relationship negatives are the hardest to synthesize and not every caption has spatial relationship descriptions.

# B   Training and Model Performance

## B.1   Model Training Setup and Hyperparameters

In the section, we present all the (hyper-)paramters we used to training TLDR model.

| Hyperparameters for training TLDR Model | |
|---|---|
| Base Model | `PaliGemma-3B-Mix-448` |
| Image Resolution | $448 \times 448$ |
| Number of Image Tokens | 1024 |
| Hidden Dimension Size | 2048 |
| LoRA Rank | 512 |
| LoRA $\alpha$ | 128 |
| LoRA dropout | 0.1 |
| GPU | $8 \times$ NVIDIA H100 |
| Batch Size | 8 |
| Gradient Accumulation Steps | 8 |
| Warmup Steps | 200 |
| Learning Rate | 0.001 |
| Learning Rate Scheduler | Cosine |

Table 8: **Hyperparameters for training TLDR Model with PaliGemma Backbone.**

| Hyperparameters for training TLDR Model | |
|---|---|
| Base Model | `Llama-3.2-11B-Vision` |
| Image Resolution | $1120 \times 1120$ |
| Number of Image Tokens | 1024 |
| Hidden Dimension Size | 4096 |
| LoRA Rank | 512 |
| LoRA $\alpha$ | 128 |
| LoRA dropout | 0.1 |
| GPU | $8 \times$ NVIDIA H100 |
| Batch Size | 8 |
| Gradient Accumulation Steps | 8 |
| Warmup Steps | 200 |
| Learning Rate | 0.001 |
| Learning Rate Scheduler | Cosine |

Table 9: **Hyperparameters for training TLDR Model with Llama Vision Backbone.**

## B.2   Performance Evaluation

In this section, we present TLDR model's performance by taxonomy, and its comparison to the naive binary RM and TLDR's ablations discussed in Section 5. We also provide the ablation study on TLDR's design choices in Table 11.

| Taxonomy | Naive RM | TLDR (ours) | ABLATION | |
| | | | Only $f_{\text{proj}}$ | Only $f_{\text{dec}}$ |
|---|---|---|---|---|
| Spatial Relationship | **74.1** | 60.2 | 50.0 (-10.2) | 51.8 (-8.4) |
| Visual Attribute | **89.8** | **89.8** | 54.7 (-35.1) | **89.8** (±0.0) |
| Attribute Binding | 88.1 | **90.6** | 55.0 (-35.6) | 85.0 (-5.6) |
| Object Identification | 73.2 | **90.6** | 51.4 (-39.2) | 86.9 (-3.7) |
| Counting | 71.0 | **73.9** | 50.7 (-23.2) | 68.1 (-5.8) |
| Small Object | **79.0** | 75.0 | 51.6 (-23.4) | 71.8 (-3.2) |
| Text OCR | 82.6 | **86.5** | 52.2 (-34.3) | 83.7 (-2.8) |
| Counterfactual | 87.3 | **90.0** | 53.3 (-36.7) | 89.3 (-0.7) |
| OVERALL | 81.1 | **83.1** | 52.5 (-30.6) | 79.4 (-3.7) |

Table 10: **Response-Level Accuracy by Taxonomy.** We find that overall TLDR models have higher response-level accuracy except for spatial relationship and small objects. We suspect the cause is from a brutal conversion of prediction from token-level to response level: by taking the product $\rho_\gamma(m, p, d) = \prod_{e \in d} \gamma(e \mid m, p, d)$, any single token $e$ has the power to veto the entire response $d$. In delicate and subtle instances such as spatial relationship and small objects, such veto power by any single token is too brutal, and a more elegant conversion from token-level probabilities $\mathbb{P}_\gamma(e \mid m, p, d)$ could be interesting for future work.

| Multimodal Projection $f_{\text{proj}}$ | Gemma Decoder $f_{\text{dec}}$ | Token-Level Accuracy $\mathcal{A}_T$ | Sentence-Level Accuracy $\mathcal{A}_S$ | Response-Level Accuracy $\mathcal{A}_R$ | mAP (neg\|pos) |
|---|---|---|---|---|---|
| ✓ | ✓ | **98.6** | **86.5** | **83.1** | (**41.3**\|**99.8**) |
| ✓ | | 97.4 | 80.0 | 52.5 | (18.2\|99.2) |
| | ✓ | 98.3 | 84.8 | 79.4 | (38.2\|99.7) |

Table 11: **Ablation Study on `PaliGemma-3B` Base Model.** It shows that unfreezing gradients on both linear projection layer $f_{\text{proj}}$ and the transformer decoder layers $f_{\text{dec}}$ are meaningful. Without tuning $f_{\text{dec}}$, the model barely works, and without $f_{\text{proj}}$, as shown in Table 10, the model is less visually grounded, especially on spatial relationship and counting.

### B.3 CORRELATION BETWEEN MODEL PERFORMANCE AND TOKEN-LEVEL HALLUCINATION RATE

This section, we draw plots to back our conjecture in Eq. 7 that, for any VLM $\xi$,

$$\text{Performance}(\xi) \propto -\log \mathcal{H}_T(\xi)$$

As shown in Figure 6, the linear correlation is very significant both visually and backed by the small $p$-value.

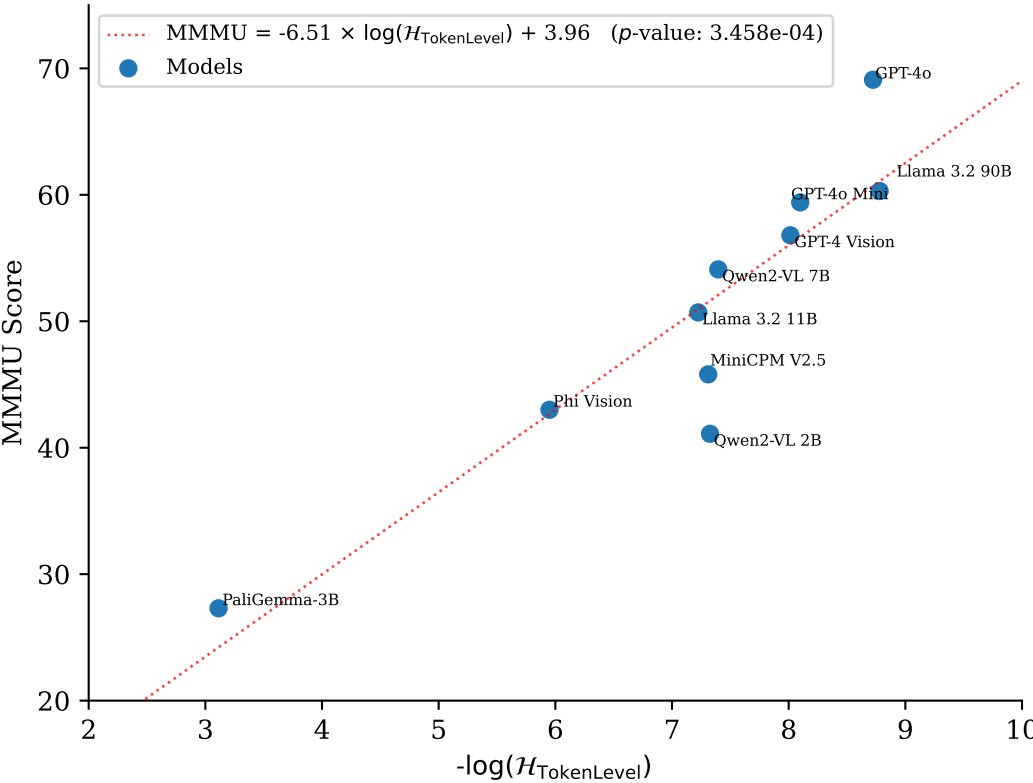

Figure 5: There is a strong linear correlation between model performance (evaluated by MMMU) and the negative log hallucination rate $-\log(\mathcal{H}_T)$, which is an approxy to model's negative log-likelihood of producing a correct token. The $p$-value of this linear correlation is 3.458e-4.

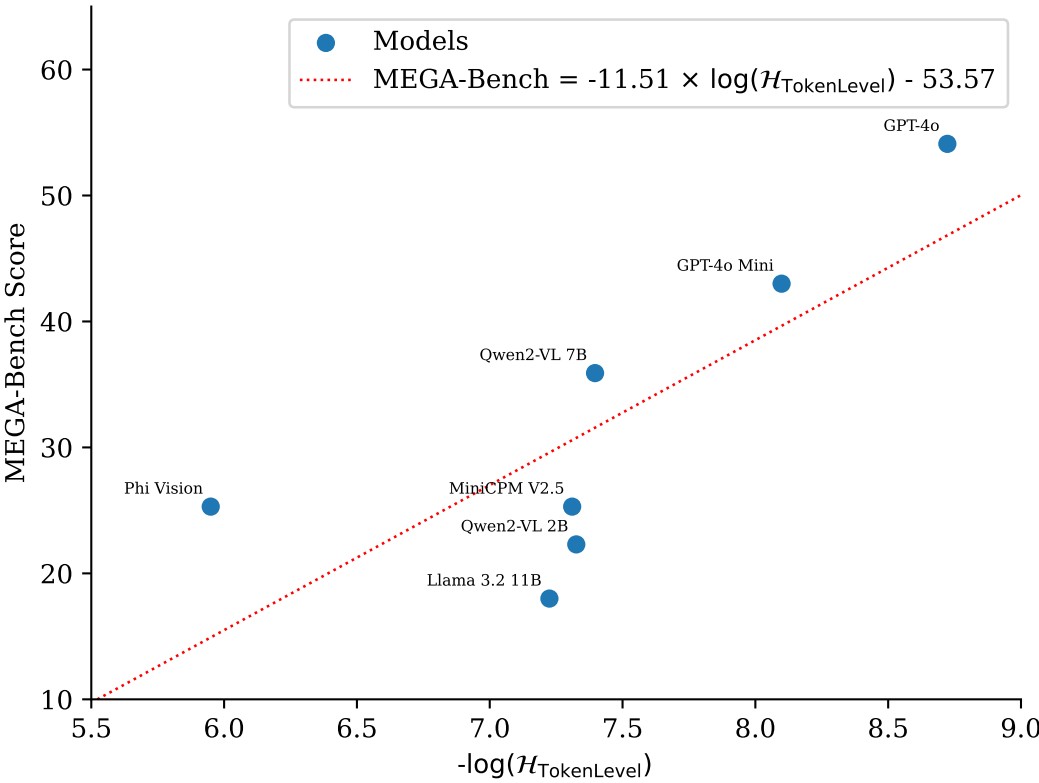

Figure 6: There is a strong linear correlation between model performance (evaluated by MEGA-Bench) and the negative log hallucination rate $-\log(\mathcal{H}_T)$, which is a proxy to model's negative log-likelihood of producing a correct token. The $p$-value of this linear correlation is 0.047.

## C  SELF CORRECTION

In the section, we present two exemplar prompts used for self-correction – one with TLDR's guidance and the other with only naive binary RM's guidance saying the original generation is wrong. The displayed image is from WinoGround.

**Note:** For better visualization, we highlight the TLDR Model's annotations in **red** when presenting the TLDR prompts.

---

**SELF CORRECTION WITH TLDR'S GUIDANCE**

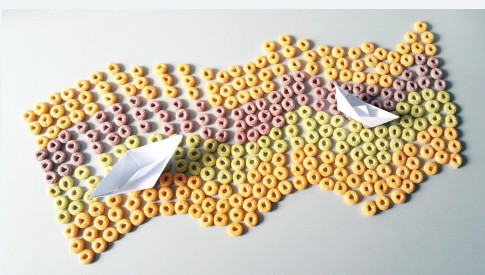

```
Image:
You task to make the provided 'Response' to the 'Text Query'
better aligned with the given image.
The 'Response' has several sentences with issues that you
need to take a closer look at.  Focus on the image more
when looking at the highlighted words or phrases in these
sentences.
Here are some rules to keep in mind:

1.  Your edited response should be as close as possible to
the original response (minimal edits) but without the errors.
2.  You should try to fix the errors by only changing
the highlighted words.  If you think these words are
hallucinated, after looking closely at the image again, you
can delete them.
3.  If you think you can't do minimal edits, you can rewrite
the whole sentences with errors.  However, do not rewrite
sentences without any annotated errors.
4.  For sentences not enumerated, you can simply copy them
if your changes won't affect them.  If you changes to other
marked sentences will also change the meaning of the unmarked
sentences, you can change them as well.  Please make the
whole paragraph coherent.
5.  The highlighted words that require extra attention are
not necessarily always incorrect.  If you think they align
well with the image, you can keep them as is.  After you
look closer to the image, use your own judgement to decide
if they need to be changed, deleted, or kept.  You may change
or delete the highlighted tokens if you think they are not
aligned with the image.  Otherwise, you can keep them as is.

Text Query:  Describe the image in details.

Response:  The image shows a creative arrangement of colorful
Lego bricks forming a shape that resembles a cat.  The bricks
are in various colors such as yellow, pink, and green.  There
are two white paper boats placed on top of the Lego cat,
one on the left and one on the right side.  The background
```

is a plain, light color, providing a neutral backdrop that
highlights the colorful **Lego cat**.

Please take a closer look at these words or phrases:  Lego,
Lego bricks, Lego cat, bricks, cat.

Now we break them into their corresponding sentences to
provide you with more context.

Please fix the sentence "The image shows a creative
arrangement of colorful Lego bricks forming a shape that
resembles a cat" with more attention to the following words:
Lego bricks, cat.

Please fix the sentence "The bricks are in various colors
such as yellow, pink, and green" with more attention to the
following words:  bricks.

Please fix the sentence "There are two white paper boats
placed on top of the Lego cat, one on the left and one on
the right side" with more attention to the following words:
Lego.

Please fix the sentence "The background is a plain, light
color, providing a neutral backdrop that highlights the
colorful Lego cat" with more attention to the following
words:  Lego cat.

Please correct all the errors.  Do not start or end the
edited response with anything extra.

Corrected Response:

---

SELF CORRECTION WITH NAIVE RM'S GUIDANCE (NO TLDR)

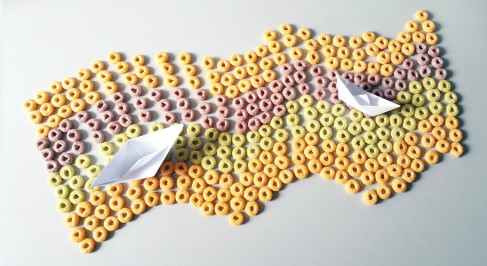

Image:
You task to make the provided 'Response' to the 'Text Query'
better aligned with the given image.
Here are some rules to keep in mind:

1.  Your edited response should be as close as possible to
the original response (minimal edits) but without the errors.
2.  If you think you can't do minimal edits, you can rewrite
the whole sentences with errors.  However, do not rewrite
sentences without any annotated errors.

Text Query:  Describe the image in details.

```
Response:  The image shows a creative arrangement of colorful
Lego bricks forming a shape that resembles a cat.  The bricks
are in various colors such as yellow, pink, and green.  There
are two white paper boats placed on top of the Lego cat,
one on the left and one on the right side.  The background
is a plain, light color, providing a neutral backdrop that
highlights the colorful Lego cat.

Please correct all the errors.  Do not start or end the
edited response with anything extra.

Corrected Response:
```

## D    LIMITATIONS

In this work, we mainly studied natural images, and whether training on natural images can extend robustly to diagrams, OCR, and more complex text-rich images is still unknown. However, we believe that as long as the training and testing distributions are similar, i.e., extending TLDR training set to text-rich images, it should also succeed. We will leave this thoughtful extension to future works. Although self-correcting smaller and less powerful models with TLDR guidance shows success, we also acknowledge the potential limitations of self-correcting more powerful models such as GPT-4o, which hallucinate less frequently, and running TLDR on each of its outputs could be costly and unrewarding. Nonetheless, it would also be nice and interesting to study if a much smaller model (e.g. Qwen2-VL-7B) with the TLDR guidance, can self-correct itself to reach GPT-4o level performance.

