# OpenReview forum: "TLDR: Token-Level Detective Reward Model for Large Vision Language Models"
_ICLR.cc/2025/Conference — ICLR 2025 Poster_

### Official Review · Reviewer_UgXE · 2024-10-18

**Soundness:** 2
**Presentation:** 3
**Contribution:** 2
**Rating:** 6
**Confidence:** 5

**Summary:**

This paper addresses the limitations of current reward models in multimodal large language models (MLLMs), which often provide only binary feedback regardless of text length. The authors propose a Token-Level Detective Reward Model (TLDR) to offer fine-grained, token-level feedback instead of coarse binary annotations. The TLDR model uses a perturbation-based method to generate hard negatives and their token-level labels for training. The paper demonstrates the benefits of TLDR models in helping models self-correct and serving as a hallucination evaluation tool. Additionally, TLDR models can accelerate human annotation by three times, improving the acquisition of high-quality vision-language data.

**Strengths:**

1 The TLDR model provides detailed token-level feedback, addressing the issue of overly simplistic binary annotations, which is a significant improvement in the evaluation process for MLLMs.

2 The perturbation-based approach for generating synthetic hard negatives adds diversity and robustness to the model training, improving the model’s ability to handle difficult scenarios.

3The model can be applied in multiple contexts, both for self-correcting MLLM outputs and as a hallucination evaluation tool, making it a versatile contribution.

**Weaknesses:**

1. Although the paper introduces the concept of token-level feedback, this approach feels like a natural extension of existing reward modeling techniques, rather than a fundamentally new innovation. The idea of fine-grained feedback has already been explored in other contexts, and the paper does not sufficiently differentiate its contribution from prior work.
2. This paper didn't compare the proposed model with existing hallucination detection and mitigation methods, such as Woodpecker or other state-of-the-art techniques. Additionally, Furthermore, the paper introduces self-correction as a novel advantage of TLDR, but other metrics and models used in hallucination detection could also be adapted to perform self-correction.
3. In Section 5.3, the authors evaluate the self-correction capabilities of the TLDR model using only one dataset, WinoGround. Relying on a single dataset raises concerns about the generalizability and reliability of the results.
4. The proposed hallucination evaluation didn’t show the superiority over the existing hallucination evaluation methods.
5.  Lack of comprehensive survey of hallucination on Large Vision-Language Models.
[1] Object hallucination in image captioning
[2] Evaluating Object Hallucination in Large Vision-Language Models
[3] FaithScore: Fine-grained Evaluations of Hallucinations in Large Vision-Language Models
[4] Analyzing and mitigating object hallucination in large vision-language models
[5] FGAIF: Aligning Large Vision-Language Models with Fine-grained AI Feedback
[6] Negative Object Presence Evaluation (NOPE) to Measure Object Hallucination in Vision-Language Models
6. Lack of evluation for the synthetic data. Meanwhile, the robustness of trained models highly rely on the quality of the synthetic data.
7. The models used to validate the SELF-CORRECTION WITH TLDR MODELS are too few, which limits the robustness of the results.

**Questions:**

See weakness

---

> ### Author Response · Authors · 2024-11-20
> **Author Response**
>
> ### W1: Although the paper introduces the concept of token-level feedback, this approach feels like a natural extension of existing reward modeling techniques, rather than a fundamentally new innovation…
>
> A1: We politely disagree that TLDR is not a new innovation. Though fine-grained feedback has been explored in NLP, the current token-level reward functions are separated models from the LLM. There is no token-level reward model for on-policy RLHF training, let alone for vision-language models. We believe TLDR establishes the stage for vision-language on-policy RLHF training. We have further clarified the differences with previous work in the revised Sec 2.
>
> ### W2, W3, W4: This paper didn't compare the proposed model with existing hallucination detection and mitigation methods, such as Woodpecker or other state-of-the-art techniques…The proposed hallucination evaluation didn’t show superiority over the existing hallucination evaluation methods.
>
> A2: Thank you for directing us to other hallucination detection and evaluation literature. The main contribution of TLDR is serving as a token-level reward model. We are excited to show the token-level reward model can also help on hallucination detection, hallucination self-correction,  and speed up human correction for the wrong words in captions. We agree it’s interesting to compare TLDR to other methods designed for hallucination mitigation. While hallucination self-correction is not our main focus, we will leave it for future study.
>
> ### W5: Lack of comprehensive survey of hallucination on Large Vision-Language Models.
>
> A5: Thank you for pointing out the missing references, we have added the listed references to the revised Sec 2 and Sec 5.3.
>
> ### W6: Lack of evaluation for the synthetic data.
>
> A6: For each taxonomy enumerated in Sec. 4, we manually inspect 30 examples with synthetic negative tokens. Taxonomy `small_object` introduces 1 example, where there was no small objects in the image, and the synthetic caption introduces an unnatural new object that is not small. Taxonomy `spatial_relationship` introduces 1 example, where the synthetic (negative) caption is a paraphrase of the original (positive) caption. All other taxonomies’ synthetic negative captions pass manual inspection that they both fit into the dedicated taxonomy and the perturbation is successful (i.e., not the paraphrase of the original caption).
>
>
> ### W7: The models used to validate the self-correction with TLDR models are too few
>
> A7: Thank you for the suggestion. We used Llama-3.2-Vision-11B and Qwen2-VL-7B as extra models for self-corrections, and we updated Table 4 in the paper as follows.
>
> | Model             | Guidance Given By | # Samples | # Samples Flagged by RM | # Self-Corrected | Win | Tie | Loss |
> |--------------------|-------------------|-----------|--------------------------|------------------|-----|-----|------|
> | GPT-4V            | TLDR (ours)       | 800       | 25                       | 21               | 12  | 7   | 2    |
> |                    | Naive             |       800    |     25                     | 15               | 2   | 11  | 2    |
> | Llama-3.2-90B     | TLDR (ours)       | 800       | 10                       | 8                | 6   | 1   | 1    |
> |                    | Naive             |     800      |             10             | 6                | 3   | 1   | 2    |
> | Qwen2-VL-7B       | TLDR (ours)       | 800       | 25                       | 16               | 9   | 5   | 2    |
> |                    | Naive             |    800       |             25             | 9                | 3   | 5   | 1    |
>
> The results validate that TLDR can assist models in better self-correcting themselves than naive binary reward models.
>
>
> Again, we are grateful for the reviewer’s thoughtful comments and suggestions. We are willing to answer any questions the reviewers may have during the discussion.

---

> > ### Comment · Reviewer_UgXE · 2024-11-22
> >
> > Thanks for your response. Your response addresses most of my concerns, but I still think your method is similar to the existing fine-grained RL method or token-level RL method, could you explain your difference and contribution in detail?

---

> > > ### Author Response · Authors · 2024-11-24
> > > **reinstating our contribution and difference in detail**
> > >
> > > We thank the reviewer for engaging in discussion with us. We would like to reinstate our contributions in detail here.
> > >
> > > **Integrated Token-Level Feedback for On-Policy RLHF**. Unlike existing token-level RL methods (such as recent works [1] [2] [3]), which typically operate as auxiliary, post-training evaluation models, TLDR is designed not only for hallucination evaluation alone but also for on-policy RLHF training. To the best of our knowledge, this is the first work to introduce token-level rewards within an on-policy RLHF framework for vision-language models. By unifying the reward model with the vision-language model, we achieve enhanced vision-language grounding capabilities and reduced hallucination rates (see Table 5 and Table 6). Furthermore, we expect TLDR to have more versatile usage than hallucination evaluation, detection, or corrections, since the synthetic training data can be easily extended to a wider amount of taxonomies, such as safety, alignments, etc., while these tasks may not fit into hallucination-focused methods’ domain.
> > >
> > > **Novel Perturbation-Based Negative Data Synthesization Pipeline for Token-Level Labels**. The perturbation-based approach we propose is novel in its ability to generate diverse, taxonomy-driven synthetic negatives. This method systematically creates challenging training scenarios for token-level reward modeling. As demonstrated in Sec. 4, the taxonomy ensures that the generated examples are interpretable and align well with the specific needs of token-level vision-language feedback, unlike prior token-level methods, which often rely on unstructured or ad-hoc negative samples.
> > >
> > > **Focus on Vision-Langue Models**. Existing token-level RL methods are predominantly designed for text-only models and are not integrated with vision-language tasks. TLDR extends the concept of token-level feedback into the multimodal domain, specifically *addressing challenges unique to vision-language models*, such as grounding issues between text tokens and visual elements, exemplified by tasks such as counting, object localization, and spatial relationships. Our results in Table 5 indicate that a simple likelihood optimization when training TLDR could improve VLMs’ performance in those categories. This is a critical advancement for aligning vision-language models with token-level supervision, which has not been explored in prior work.
> > >
> > > **Practical Applications in Vision-Language Settings**. TLDR’s contributions extend beyond hallucination detection, serving as a multi-purpose tool for:
> > > Improving the model’s alignment with an implicit Likelihood Optimization for free
> > > Efficient self-correction to refine outputs.
> > > Accelerated human annotation workflows by pre-annotating token-level issues. This gives rise to the potential of collecting more high-quality human-annotated vision language data.
> > >
> > > We again thank the reviewer for the discussion with us. Feel free to post any further questions and concerns.
> > >
> > > [1] Yong et al. TLCR: Token-Level Continuous Reward for Fine-grained Reinforcement Learning from Human Feedback, 2024.
> > >
> > > [2] Yang et al. Selective Preference Optimization via Token-Level Reward Function Estimation, 2024.
> > >
> > > [3] Zeng et al. Token-level Direct Preference Optimization, 2024.

---

> > > ### Author Response · Authors · 2024-12-02
> > >
> > > Dear Reviewer UgXE,
> > >
> > > As the discussion period is about to end, please let us know if our response has resolved your concerns. If you have further questions or concerns, please let us know as well. Thank you participating the discussion.
> > >
> > > Best,
> > > Authors of TLDR

---

> > > > ### Comment · Reviewer_UgXE · 2024-12-03
> > > >
> > > > Thanks for your response. I increase my score due to authors addressed most of my concerns.

---

### Official Review · Reviewer_f5mm · 2024-10-27

**Soundness:** 3
**Presentation:** 3
**Contribution:** 3
**Rating:** 6
**Confidence:** 4

**Summary:**

This paper presents a token-level reward model for large vision-language models, where the reward model would produce reward labels for each token during the generation.

To obtain a training dataset for the reward model, this paper proposes to perturb image caption/VQA answer with an LLM (i.e., LLaMA-70B), with specific templates defined to generate more hallucination/bottleneck-oriented perturbations (e.g., counting objects, color identification etc.)

After training with the dataset, a base LVLM becomes a Token-Level Reward Model for various applications:

(i) hallucination evaluation: where the reward labels could be used to develop token-, sentenc- and response-level accuracy for the generated responses.

(ii) self-correction with the token-level reward for reducing hallucination;

(iii) enhanced performance as a by-product of the reward model training;

(iv) speed-up the caption annotation

**Strengths:**

- Overall, I think the idea of developing a token-level reward model for LVLMs is interesting and the authors propose a feasible dataset synthesis method to achieve this.
- The paper is well-written and well-organized.
- The applications of TLDR seem to be promising for future LVLM developments.

**Weaknesses:**

My major concerns lie in the experimental settings and evaluation setups, which makes the results less convincing to me.

- Backbone choices: I found the models used in this paper are somewhat arbitrary: the reward model is trained upon PaliGemma; human evaluation is conducted with captions generated by MiniCPM; GPT-4V is used to perform self-correction experiments. Are there any specific reasons to choose different models in these experiments?

- Insufficient experiments regarding model selection:
   - The reward model is only trained with a PaliGemma-3B model, which as demonstrated in Table 3, performs the worst in the hallucination. Wouldn't a stronger backbone lead to stronger Token-level reward models? An ablation study comparing TLDR models trained on different backbones, including stronger ones like phi and llama-3.2, is recommended.
   - The correlation between MMMU scores is misleading. As pointed by Cambrian-1 and MMMU-Pro, MMMU score does not faithfully reflect the visual understanding and reasoning performance but more about the LLM capability. This renders the correlation analysis less convincing and I recommend the authors incorporate visual-oriented benchmarks to justify this claim.
   - Why LLaVA-series and Qwen-VL are not included in the evaluation? These are commonly adopted LVLMs.  Including these models would provide a more comprehensive comparison across different LVLM architectures and training approaches.


- In Table 5, why are only two tasks (Counting and spatial relation) of BLINK adopted as in-domain tasks? There are also in-domain  tasks such as Visual correspondence & Object localization. Adding the comprehensive results on BLINK could better illustrate the full picture.

**Questions:**

- Could you explain their rationale for choosing different models for each experiment?



Minor:
There are empty lines after Eq. 6 and Eq. 7.

This paper ignores many relevant papers regarding LVLMs alignments and reward modeling. It would be more beneficial to compare the reward models or DPO training performance of these studies:

- LLaVA-RLHF: Aligning Large Multimodal Models with Factually Augmented RLHF
- RLAIF-V: Aligning MLLMs through Open-Source AI Feedback for Super GPT-4V Trustworthiness
- Rlhf-v: Towards trustworthy mllms via behavior alignment from fine-grained correctional human feedback
- VLFeedback: A Large-Scale AI Feedback Dataset for Large Vision-Language Models Alignment
-  Strengthening Multimodal Large Language Model with Bootstrapped Preference Optimization

---

> ### Author Response · Authors · 2024-11-20
> **Author Response**
>
> We are happy to see that the reviewer found our paper well-written and organized and appreciated the idea of developing a token-level reward model for LVLMs. We are also pleased that they recognized the potential applications of our model, including hallucination evaluation, self-correction, enhanced model performance, and its ability to significantly speed up caption annotation.
>
> ## W1, W2: Backbone choices for TLDR
> We want to have a lightweight reward model that can provide meaningful feedback for larger models, and PaliGemma's 3B size is a nice fit. Llama-3.2-Vision models release date (Sep. 25) was too close to the conference submission deadline (Oct. 1). During the rebuttal phase, in order to showcase the effectiveness of TLDR training, we ran experiments using Llama-3.2-Vision-11B as the backbone model for TLDR, and here are the llama-only results, with the full results updated in the revision and the global response.
>
> *Table 1: Evaluation of Model Performance*
> | Base Model            | Reward Model | Token-Level Accuracy $A_T$ | Sentence-Level Accuracy $A_S$ | Response-Level Accuracy $A_R$ | mAP (neg,pos)   |
> |-----------------------|--------------|-------------------------------|----------------------------------|----------------------------------|-----------------|
> | Llama-3.2-11B-Vision | TLDR         | 98.9                         | 90.8                            | 88.2                            | (45.7,99.8)     |
> |                       | Naive        | —                             | —                                | 86.7                            | —               |
>
> *Table 5: TLDR automatically performs likelihood optimization*
> | Models                    | BLINK ↑                 |                       |                  | IsoBench ↑         |                  |
> |---------------------------|-------------------------|-----------------------|------------------|--------------------|------------------|
> |                           | Count                  | Spatial Relation      | Object Localize  | Function Parity    | Chess Winner     |
> | Llama-3.2-11B-Vision      | 55.0                   | 61.5                 | 60.7             | 34.9               | 45.5             |
> | + TLDR ($ \tau = 0.25 $) | **67.5**               | 65.0                 | **67.2**         | **35.4**           | 43.6             |
> | + TLDR ($ \tau = 0.5 $) | 65.8                   | **65.7**             | 59.0             | 33.3               | 43.7             |
> | + TLDR ($ \tau = 1 $)    | 61.7                   | **65.7**             | 56.6             | 35.1               | 39.4             |
>
>
> ## W1: Model choices for human evaluation and self-correction
> **Why we chose miniCPM**: We want to have more samples where the model hallucinates. For bigger models like GPT-4o, the hallucination rate is too low to draw significant amount of samples. so we used captions generated by 7B models such as miniCPM. we conducted extra experiments with Phi-Vision and Qwen2-VL-7B, with a special focus on false negative (FN) type of errors, and averaged among three human annotators, we find the TLDR model has a sentence-level FN rate of 8.7%, 10.5% and 9.8%, on MiniCPM, Phi-Vision and Qwen2-VL-7B respectively.
>
> **Why we chose GPT-4V for self-correction**: Most open-weight multimodal models still suffer from instruction following, such as correcting the captions. In this sense, we need a model that is able to self-correct provided with some signals. We didn’t choose GPT-4o because, again, it rarely hallucinates. To make the self-correction experiments more convincing, we apply self-correction with TLDR to two new models, Llama-3.2-90B-Vision and Qwen2-VL-7B, both of which can self-correct given some signals. Here are the results of them
>
> | Model             | Guidance Given By | # Samples | # Samples Flagged by RM | # Self-Corrected | Win | Tie | Loss |
> |--------------------|-------------------|-----------|--------------------------|------------------|-----|-----|------|
> | Llama-3.2-90B     | TLDR (ours)       | 800       | 10                       | 8                | 6   | 1   | 1    |
> |                    | Naive             |     800      |             10             | 6                | 3   | 1   | 2    |
> | Qwen2-VL-7B       | TLDR (ours)       | 800       | 25                       | 16               | 9   | 5   | 2    |
> |                    | Naive             |    800       |             25             | 9                | 3   | 5   | 1    |
>
> We still find TLDR can assist these two models in better self-correcting themselves than naive binary reward models.

---

> > ### Author Response · Authors · 2024-11-20
> > **Author Response (contd.)**
> >
> > ## W3: Correlation between MMMU scores and Token-Level Hallucination Rates
> > we first highlight our person R correlation test in the caption of Table 3: the correlation score between $−\ log (\mathcal H_T)$ and MMMU score is 0.902 with a p-value of 3.45e-4, where $\mathcal H_T$ is the token-level hallucination rates. We also provide a new Fig. 5 in the appendix to show this linear trend that, given the models we evaluated (including newly added Qwen2-VL), $\mathrm{MMMU} = -6.51 \times \log(\mathcal H_T) + 3.96$.
> >
> > We chose MMMU since it covers the widest range of models. We also include MEGA-bench [1] in the updated Table 3 and the trend between MEGA-bench score and $−\ log (\mathcal H_T)$ is also linear with a correlation coefficient of 0.759 with a p-value of 0.047. We include the plots for for both MMMU and MEGA-Bench in Figures 5 and 6 respectively.
> >
> >
> > ## W4: Evaluating Llava and Qwen-VL
> > We decided to perform an extra evaluation on the Qwen2-VL family with 2B and 7B versions. We decided not to evaluate the Llava series since their training data were derived from GPT-generated texts, which is a clear violation of OpenAI’s terms of service. We updated Table 3 in the paper and here are the results for Qwen2-VL’s,
> >
> >
> > | Models                     | Hallucination Rate (%) ↓ |                       |                   | MMMU ↑ | MEGA-Bench ↑ |
> > |----------------------------|--------------------------|-----------------------|-------------------|--------|--------------|
> > |                            | Token-Level             | Sentence-Level       | Response-Level    |        |              |
> > | Qwen2-VL-7B                | 0.061                   | 0.48                 | 1.96              | 54.1   | 35.9         |
> > | Qwen2-VL-2B                | 0.066                   | 0.72                 | 1.70              | 41.1   | 22.3         |
> >
> >
> > ## W5: More tasks from BLINK
> > We thank the reviewer for the suggestion. Since the suggested visual correspondence task is multi-image, which is not naturally supported by many models, we conducted extra evaluations on the Object Localization task, together with our new Llama-3.2 backed TLDRs. The full table is updated in the paper revision and in the global response. We highlight the Object Localization task here.
> >
> > | Models                    | Object Localize  |
> > |---------------------------|------------------|
> > | PaliGemma-3B              | 45.9             |
> > | + TLDR ($ \tau = 0.25 $)  | **47.5**         |
> > | + TLDR ($ \tau = 0.5 $)   | 42.6             |
> > | + TLDR ($ \tau = 1 $)     | 42.6             |
> > | Llama-3.2-11B-Vision      | 60.7             |
> > | + TLDR ($ \tau = 0.25 $)  | **67.2**         |
> > | + TLDR ($ \tau = 0.5 $)   | 59.0             |
> > | + TLDR ($ \tau = 1 $)     | 56.6             |
> >
> > ## Q1: Extra lines in Eqn 6 and 7
> > Thank you for pointing them out. We have fixed them.
> >
> > ## Q2: Missing references
> > We thank the reviewer for the suggested related work. We will have a more comprehensive related work section including *all* the suggested work here in our next revision.
> >
> > Again, we are grateful for the reviewer’s thoughtful comments and suggestions. We are willing to answer any questions the reviewers may have during the discussion.

---

> > > ### Comment · Reviewer_f5mm · 2024-11-22
> > >
> > > Thank you for your response and clarifications.
> > >
> > > ## RE: W1, W2: Backbone choices for TLDR
> > >
> > > I understood that a lightweight RM might be more efficient for guiding larger models, and the additional LLaMa-3.2 11B results are appreciated. Given that Qwen2-VL-2B is even smaller, I would suggest the authors add this model as well in the next revision.
> > >
> > > ## RE: W1: Model choices for human evaluation and self-correction
> > >
> > > The motivation behind these choices is better elaborated given the clarifications.
> > > However, this triggers a question that the proposed model cannot guide SOTA models such as 4o, restricting the potential application scenarios (e.g., you cannot guide GPT-4o with Paligemma). Adding this to limitations would be a good choice.
> > >
> > > ## RE: W3: Correlation between MMMU scores and Token-Level Hallucination Rates
> > > The newly added MEGA-bench correlation looks good to me. A sanity-check question is:
> > > What's the GPT-4o version you are using? Is consistent with the model used by MMMU-Pro and MEGA-bench?
> > >
> > > ## RE: W4 and W5
> > > Thanks for these additional results, which would be a plus for the manuscript.
> > >
> > > ## RE: Q2
> > >
> > > In the revised PDF, the references suggested by Reviewer UgXE are incorporated, while my suggestions are not added as promised :-(
> > >
> > > Given most my concerns are resolved, I will increase my scores accordingly.

---

> > > > ### Author Response · Authors · 2024-11-22
> > > >
> > > > We are glad to here that most of your concerns are resolved and you're willing to increase the score.
> > > >
> > > > ### Alternative Backbone
> > > > Thanks for the suggestion, and we will definitely repeat the same experiments on Qwen2-VL-2B as the much stronger and lightweight model than PaliGemma-3B.
> > > >
> > > > ## Limitation of Self-Correction
> > > > Yes. We'll add this to the limitations that models like 4o, which hallucinates less frequently, may not need self-corrections. On the other hand, it would also be nice and interesting to study if a much smaller model (e.g. Qwen2-VL-7B) with the TLDR guidance, can self-correct itself to reach GPT-4o level performance. We'll leave this for future study.
> > > >
> > > > ## GPT-4o versions for MMMU and MEGA-Bench
> > > > We are using the reported numbers in the MMMU leaderboard and MEGA-Bench leaderboard. I checked the versions on the leaderboards and they are both GPT-4o (0513).
> > > >
> > > > ## Missing reference
> > > > Apologies for a slight rush for posting the rebuttal. Your suggested references are very meaningful and we are still working on the revision to have a detailed discussion. We will update it all together with the new limitation section soon during this weekend.
> > > >
> > > > Again, we appreciate your discussion with us, and the comments, suggestions, and questions are very insightful. Thank you very much! Please let us know if you have more questions.

---

### Official Review · Reviewer_KRLQ · 2024-11-04

**Soundness:** 3
**Presentation:** 4
**Contribution:** 3
**Rating:** 6
**Confidence:** 4

**Summary:**

This paper presents a token-level reward model for VLMs. The challenge it aims to address is that the commonly used binary reward is often biased towards linger captions. The proposed TLDR model can provide per-token feedback and solve issues such as hallucinations. The method achieves this using synthetic data generation by perturbing correct captions.

**Strengths:**

- The paper is well motivated
- The method section is clear and easy to follow
- Some very interesting findings, such as that fine-tuning a VLM for token-level rewards improves the model itself.

**Weaknesses:**

- I would not really call TLDR a reward model, as the paper has not shown that it can actually be used as a reward model (in the RLHF sense). In its current form, this is a per-token correctness model.
- Table 9 shows that in the response-level evaluation, TLDR is only marginally better that a naive response-revel reward model. While this is not possible for the response-level model, it would be interesting to see if TLDR outputs the correct prediction for the right reason, i.e. manually evaluate per-token accuracy, recall and precision, rather than just the global metrics.
- Is Equation (8) backed by anything? That seems very arbitrary.

**Questions:**

-

---

> ### Author Response · Authors · 2024-11-20
> **Author Response**
>
> We thank the reviewers for their thoughtful comments and suggestions. We are happy to see that the reviewer appreciated the clear motivation behind our work and recognized the innovation in shifting to token-level feedback to address the limitations of binary annotations. We are also pleased that the clarity of our method section was acknowledged, as well as the interesting finding that fine-tuning a vision-language model with token-level rewards improves its overall performance.
>
> ## W1: Whether TLDR can be called a reward model
> In the context of PPO, a reward model refers to a separate machine learning model that estimates a numerical reward score for an agent's action or output, based on how well it aligns with the desired behavior. In short, we believe that as long as a separate model can provide positive and negative feedback, that is aligned with a desired behavior, it can be called a reward model. In our context, TLDR resembles the reward model setting for PPO training while giving token-level rewards. Although we didn’t perform large-scale RLHF, which could be very costly, we provide an approxy in Sec 5.4, whereas training the TLDR model secretly tunes the backbone models to align with the reward model head – which we call a likelihood optimization method. We believe this oversimplified trick is also an RLHF procedure that is less costly and could improve model performance, as shown in Tables 5 and 6. We also believe TLDR shed lights for future work in developing a complex and effective system using token-level rewards.
>
> ## W2: Need human annotations for model quality.
> We manually examined 100 Phi-vision generated caption sentences with at least one “bad” tag, only 10.5% of the sentences are actually correct (i.e., with at least one “bad” tag wrongly marked). Similarly, we conducted the human evaluations TLDR’s detection on Qwen2-VL-7B generated captions as well, and the false negative rate there is 9.8%.
>
> ## W3: Evidence for Equation (8)
> We highlight our person R correlation test in the caption of Table 3: the correlation score between $−\ log (\mathcal H_T)$ and MMMU score is 0.902 with a p-value of 3.45e-4, where $\mathcal H_T$ is the token-level hallucination rates. We also provide a new Fig. 5 in the appendix to show this linear trend that, given the models we evaluated (including newly added Qwen2-VL), $\mathrm{MMMU} = -6.51 \times \log(\mathcal H_T) + 3.96$. Such empirical finding induces our conjecture in Equation (8).
>
> Again, we are grateful for the reviewer’s thoughtful comments and suggestions. We are willing to answer any questions the reviewers may have during the discussion.

---

> ### Author Response · Authors · 2024-11-28
> **Reminder for discussion**
>
> Dear Reviewer KRLQ,
>
> We would like to kindly remind you to take a look at our rebuttal and we are willing to discuss further to address any questions or concerns you may have. Thanks!
>
> Best,
> Authors of TLDR

---

### Official Review · Reviewer_PRNS · 2024-11-04

**Soundness:** 3
**Presentation:** 3
**Contribution:** 3
**Rating:** 6
**Confidence:** 3

**Summary:**

The paper introduces a model that addresses the limitations of existing reward models, which provide only binary feedback for entire text sequences. Instead, it assigns rewards at each token, enabling more precise and interpretable feedback, improving self-correction, hallucination detection, and human annotation efficiency in vision-language tasks.

**Strengths:**

1. The idea is interesting, as it shifts from traditional binary feedback to a more detailed token-level approach.
2. The model provides token-level rewards, offering more precise and interpretable feedback compared to traditional binary reward models.
3. The proposed HALLUCINATION RATE (%) is a novel evaluation metric.
4. The token-level errors can guide multimodal large language models in self-correction, enhancing their performance.
5. The model can serve as a data correction tool, effectively speeding up the process by three times.

**Weaknesses:**

The model is trained using a perturbation-based data generation process based on simple factual statements, which introduces some limitations:
1. Its performance may be limited when interpreting information-rich images like posters, where elements for the same noun are not unique and are arranged in a complicated layout.
2. The model may also face challenges in understanding text-rich images, such as documents where relationships between concepts are described in text and require logical reasoning.

**Questions:**

1. In Section 5.4, despite the model’s low performance on tasks listed in Table 5, does the model with ($\tau$ = 1) still achieve the best performance on token-level hallucination detection?
2. Do you have any insights into why the finetuned LoRA weights initially enhance the model’s performance on other tasks but then cause a decline as ($\tau$) increases?

---

> ### Author Response · Authors · 2024-11-20
> **Author Response**
>
> We thank the reviewer for their comments, kind words, and recognition. We are glad that the reviewer praised the shift from binary to token-level rewards for providing more precise and interpretable feedback, highlighted the novel hallucination rate metric, and noted the model’s utility in improving self-correction and speeding up data annotation. We are also happy to see that the reviewer shares the same insights with us about TLDR’s potential as a data correction tool.
>
> ## Concerns about limitations in interpreting complex, text-rich images
> In this work, we mainly studied *natural images*, and whether training on natural images can extend robustly to diagrams, OCR, and more complex text-rich images is still unknown. However, we believe that as long as the training and testing distributions are similar, i.e., extending TLDR training set to text-rich images, it should also succeed. We will leave this thoughtful extension to future works, and we thank the reviewer for pointing this out.
>
> ## Effect of $\tau$
> When $\tau = 1$, the LoRA $\alpha$’s for finetuning TLDR and running inference with TLDR backbone model are the same, so in this case, it’s the finetuned TLDR reward model, and its performance is shown in Table 1. Inspired by the reviewer’s question,  we conducted extra experiments to show that backbone models, after tuned with TLDR, can reduce their hallucination rates significantly. We created a new Table 6 in the paper as follows,
>
> | Models                    | Hallucination Rate (%) ↓ |                       |                   |
> |---------------------------|--------------------------|-----------------------|-------------------|
> |                           | Token-Level             | Sentence-Level       | Response-Level    |
> | PaliGemma-3B              | 4.444                   | 5.96                 | 17.50             |
> | + TLDR ($ \tau = 0.10 $) | 0.991                   | 3.80                 | 10.53             |
> | + TLDR ($ \tau = 0.25 $) | **0.172**               | **1.13**             | **3.96**          |
> | Llama-3.2-11B-Vision      | 0.073                   | 0.85                 | 1.88              |
> | + TLDR ($ \tau = 0.10 $) | 0.078                   | **0.69**                 | 2.71              |
> | + TLDR ($ \tau = 0.25 $) | **0.066**               | 0.74            | **1.72**          |
>
> The reviewer also asked why when $\tau$ increases beyond, for example, 0.25, the model performance decreases. Let’s take one step back to understand what $\tau$ represents. When $\tau = 0$, the merged model is the original backbone model (PaliGemma or Llama-3.2-Vision), while when $\tau = 1$, the merged model is the finetuned TLDR model’s backbone, which is optimized to provide token-level *rewards*. In this regard, $\tau$ can be viewed as the interpolation coefficient between a vision language model ($\tau=0$) and a reward model ($\tau = 1$). When $\tau$ is chosen as a good midpoint, it takes the benefit of both – can generate fluent and meaningful text, and can hallucinate less. When $\tau$ increases beyond, it shifts its behavior towards being a reward model, and its language abilities could decrease, resulting in lower scores (blink, isobench, and hallucination rates) that are calculated from text generations.
>
> Again, we are grateful for the reviewer’s thoughtful comments and suggestions. We are willing to answer any questions the reviewers may have during the discussion.

---

> ### Author Response · Authors · 2024-11-28
> **Reminder for discussion**
>
> Dear Reviewer PRNS,
>
> We would like to kindly remind you to take a look at our rebuttal and we are willing to discuss further to address any questions or concerns you may have. Thanks!
>
> Best,
>
> Authors of TLDR

---

> > ### Comment · Reviewer_PRNS · 2024-12-03
> >
> > Thank you for the detailed response. Most of my concerns have been adequately addressed, and I would like to maintain my positive rating.

---

### Author Response · Authors · 2024-11-20
**Global Response -- Thank Reviewers and Reinstate Our Contribution**

We first thank the reviewers for their thoughtful suggestions and comments. We appreciate the reviewers’ kind words in acknowledging our contributions. The reviewers appreciated the **clear motivation and novelty** of introducing token-level feedback for vision-language models, addressing limitations of binary annotations with **more precise and interpretable rewards** (KRLQ, UgXE, PRNS). They commended the paper’s **clarity and organization** (KRLQ, f5mm) and recognized the potential applications of TLDR in **hallucination evaluation, self-correction, and accelerating human annotation** (f5mm, UgXE, PRNS). The **synthetic data generation** approach was praised for improving robustness and diversity (UgXE), and the **performance improvements from fine-tuning** with token-level rewards were highlighted as an interesting finding (KRLQ).

We would like to reiterate our main contributions here:

- **Novel Token-Level Detective Reward (TLDR) Model**: We propose a novel unified TLDR model for vision-language tasks, offering a binary reward at the token level. TLDR establishes the stage for vision-language on-policy RLHF training with token-level reward.

- **Hallucination Detection and Correction**: Our reward model TLDR effectively enables the detection and correction of hallucinations, significantly accelerating human intervention for correcting “hallucinated” words.

- **Improved Downstream Vision-Language Performance**: By unifying the reward model with the vision-language model, we achieve enhanced vision-language grounding capabilities and reduced hallucination rates.

---

### Author Response · Authors · 2024-11-20
**Global Response -- Additional Experiments**

We conducted extra experiments as recommended by reviewers. Notably, we extend TLDR’s method beyond a single backbone model, PaliGemma-3B, by repeating the same procedure on the newly released Llama-3.2-11B-Vision.

We updated our manuscript highlighting all changes and extra experiments in red texts or shades.

## Alternative Backbone Model for TLDR
As reviewer f5mm suggested, we conducted extra experiments on alternative backbone models for TLDR. Many reviewers may share the same concern. Now, we choose Llama-3.2-11B-Vision-Instruct as the new backbone model, and we have updated Table 1 in the revision as such

| Base Model            | Reward Model | Token-Level Accuracy  | Sentence-Level Accuracy | Response-Level Accuracy  | mAP (neg,pos)   |
|-----------------------|--------------|-------------------------------|----------------------------------|----------------------------------|-----------------|
| PaliGemma-3B         | TLDR         | 98.6                         | 86.5                            | **83.1**                            | (**41.3**,99.8)     |
|   PaliGemma-3B    | Naive        | —                             | —                                | 81.1                            | —               |
| Llama-3.2-11B-Vision | TLDR         | 98.9                         | 90.8                            | **88.2**                            | (**45.7**,99.8)     |
|  Llama-3.2-11B-Vision   | Naive        | —                             | —                                | 86.7                            | —               |
| GPT-4o               | Prompting    | 95.5                         | 66.9                            | 52.9                            | (19.7,98.1)     |
| Random Guess         | —            | —                             | —                                | 50.0                            | —               |

We also include a zero-shot GPT-4o baseline to demonstrate that simply prompting multimodal LLMs won’t give comparable performance to TLDR with much smaller models. This poses one explanation of why multimodal LLMs can't perform self-correction well with only binary feedbacks for the entire response and why token-level feedbacks are necessary.

## Updated TLDR Likelihood Optimization Results
As we claimed in Sec. 5.4, when training the reward model head for TLDR, it automatically optimizes the backbone model in the form of “likelihood optimization” for each token. Inspired by Reviewer PRNS, we conducted extra experiments to show that backbone models, after tuned with TLDR, can reduce their hallucination rates significantly. We hope these results can clear some concerns by Reviewer KRLQ by showing TLDR is actually a simplified token-level reward model, and the likelihood optimization for free is also a simplified RLHF process. We created a new Table 6 in the paper as follows,

| Models                    | Hallucination Rate (%) ↓ |                       |                   |
|---------------------------|--------------------------|-----------------------|-------------------|
|                           | Token-Level             | Sentence-Level       | Response-Level    |
| PaliGemma-3B              | 4.444                   | 5.96                 | 17.50             |
| + TLDR ($ \tau = 0.10 $) | 0.991                   | 3.80                 | 10.53             |
| + TLDR ($ \tau = 0.25 $) | **0.172**               | **1.13**             | **3.96**          |
| Llama-3.2-11B-Vision      | 0.073                   | 0.85                 | 1.88              |
| + TLDR ($ \tau = 0.10 $) | 0.078                   | **0.69**                 | 2.71              |
| + TLDR ($ \tau = 0.25 $) | **0.066**               | 0.74             | **1.72**          |

---

> ### Author Response · Authors · 2024-11-20
> **Additional Experiments (contd.)**
>
> Reviewer f5mm also suggested more experiments with BLINK. Since the suggested visual correspondence task is multi-image, which is not naturally supported by many models, we conducted extra evaluations on the Object Localization task, together with our new Llama-3.2 backed TLDRs. We updated Table 5 as such
>
> | Models                    | BLINK ↑                 |                       |                  | IsoBench ↑         |                  |
> |---------------------------|-------------------------|-----------------------|------------------|--------------------|------------------|
> |                           | Count                  | Spatial Relation      | Object Localize  | Function Parity    | Chess Winner     |
> | PaliGemma-3B              | 69.2                   | 78.3                 | 45.9             | 41.4               | 45.1             |
> | + TLDR ($ \tau = 0.25 $) | **71.7**               | 80.4                 | **47.5**         | **45.1**           | **47.5**         |
> | + TLDR ($ \tau = 0.5 $) | **71.7**               | **81.1**             | 42.6             | 44.3               | **47.5**         |
> | + TLDR ($ \tau = 1 $)    | 12.5                   | 2.1                  | 42.6             | 34.4               | 44.4             |
> | Llama-3.2-11B-Vision      | 55.0                   | 61.5                 | 60.7             | 34.9               | 45.5             |
> | + TLDR ($ \tau = 0.25 $) | **67.5**               | 65.0                 | **67.2**         | **35.4**           | 43.6             |
> | + TLDR ($ \tau = 0.5 $) | 65.8                   | **65.7**             | 59.0             | 33.3               | 43.7             |
> | + TLDR ($ \tau = 1 $)    | 61.7                   | **65.7**             | 56.6             | 35.1               | 39.4             |
>
> We find that the TLDR model with both PaliGemma and Llama-3.2 backbone can be improved with the token-level reward training for free.
>
>
> ## Updated Hallucination Evaluation Benchmark Table
> As reviewer f5mm also suggested, we should evaluate the Llava series and Qwen VL series. We decided to perform an extra evaluation on the Qwen2-VL family with 2B and 7B versions. We decided not to evaluate the Llava series since their training data were derived from GPT-generated texts, which is a clear violation of OpenAI’s terms of service. We updated Table 3 in the paper as follows,
>
> | Models                     | Hallucination Rate (%) ↓ |                       |                   | MMMU ↑ | MEGA-Bench ↑ |
> |----------------------------|--------------------------|-----------------------|-------------------|--------|--------------|
> |                            | Token-Level             | Sentence-Level       | Response-Level    |        |              |
> | GPT-4o                     | **0.016**               | 0.23                 | 1.62              | **69.1** | **54.1**       |
> | Llama-3.2-90B-Vision       | 0.017               | **0.19**             | **1.23**          | 60.3   | 43.0       |
> | GPT-4o-mini                | 0.030                   | 0.38                 | 2.12              | 59.4   | /            |
> | GPT-4-Turbo-Vision         | 0.033                   | 0.62                 | 3.12              | 56.8   | /            |
> | Qwen2-VL-7B                | 0.061                   | 0.48                 | 1.96              | 54.1   | 35.9         |
> | Qwen2-VL-2B                | 0.066                   | 0.72                 | 1.70              | 41.1   | 22.3         |
> | MiniCPM-Llama-3-V2.5       | 0.067                   | 0.81                 | 3.62              | 45.8   | 22.8         |
> | Llama-3.2-11B-Vision       | 0.073                   | 0.85                 | 1.88              | 50.7   | 18.0         |
> | Phi-Vision-3.5-Instruct    | 0.261                   | 2.65                 | 9.25              | 43.0   | 25.3         |
> | PaliGemma-3B-Mix-448       | 4.444                   | 5.96                 | 17.50             | 27.3   | /            |
>
> As f5mm also pointed out, MMMU scores may not be fully representative of model performance. We chose MMMU as the score to compare because it has the most coverage of models so far. We add extra scores from MEGA-Bench [1]. As reviewer KRLQ suggested we should provide more evidence for Eqn. (8), and we first highlight our person R correlation test in the caption of Table 3: the correlation score between $−\ log (\mathcal H_T)$ and MMMU score is 0.902 with a p-value of 3.45e-4, where $\mathcal H_T$ is the token-level hallucination rates. We also provide a new Fig. 5 in the appendix to show this linear trend that, given the models we evaluated (including newly added Qwen2-VL), $\mathrm{MMMU} = -6.51 \times \log(\mathcal H_T) + 3.96$.
>
> [1] Chen, Jiacheng, et al. "MEGA-Bench: Scaling Multimodal Evaluation to over 500 Real-World Tasks." arXiv preprint arXiv:2410.10563 (2024).

---

> > ### Author Response · Authors · 2024-11-20
> > **Additional Experiments (contd.)**
> >
> > ## Human Annotation
> > ### Human Annotation for Evaluating Model Performance
> > As discussed in Sec. 5.1, we have human evaluations on 100 TLDR’s predictions on MiniCPM-Llama3.2’s captioning responses, and find TLDR’s sentence-level false negative rates (i.e., the frequency at which the original sentence is correct but TLDR labels some tokens in this sentence as wrong tokens.) of 8.7%. Reviewer f5mm suggested this might not be comprehensive since it’s only based on MiniCPM’s outputs. During the rebuttal period, we conducted extra human annotations on 100 TLDR’s prediction on Phi-Vision’s outputs and 100 on Qwen2-VL-7B’s outputs and calculated TLDR’s false negative rates — 10.5% on Phi-Vision outputs and 9.8% on Qwen2-VL-7B’s outputs.
> >
> > ### Human Annotation for Self-Correction with More Models
> > As Reviewer UgXE suggested, we conduct self-correction with TLDR on more models than just GPT-4V. During the rebuttal period, we used Llama-3.2-Vision-11B and Qwen2-VL-7B as extra models for self-corrections, and we updated Table 4 in the paper as follows
> >
> > | Model             | Guidance Given By | # Samples | # Samples Flagged by RM | # Self-Corrected | Win | Tie | Loss |
> > |--------------------|-------------------|-----------|--------------------------|------------------|-----|-----|------|
> > | GPT-4V            | TLDR (ours)       | 800       | 25                       | 21               | 12  | 7   | 2    |
> > |                    | Naive             |       800    |     25                     | 15               | 2   | 11  | 2    |
> > | Llama-3.2-90B     | TLDR (ours)       | 800       | 10                       | 8                | 6   | 1   | 1    |
> > |                    | Naive             |     800      |             10             | 6                | 3   | 1   | 2    |
> > | Qwen2-VL-7B       | TLDR (ours)       | 800       | 25                       | 16               | 9   | 5   | 2    |
> > |                    | Naive             |    800       |             25             | 9                | 3   | 5   | 1    |
> >
> > We find TLDR can assist models in better self-correcting themselves than naive binary reward models.

---

### Meta-Review · Area_Chair_Qm3S · 2024-12-19

**Metareview:**

All reviewers gave positive scores. The authors introduce a token-level detective reward model for large vision-language models, addressing the limitations of existing binary reward models by providing fine-grained, token-level feedback. This work could benefit the community well by enhancing the development and evaluation of vision-language models. The authors should still further revise the paper to make it easier to follow. Some parts, such as the detailed contributions and experimental setup, should be further clarified to enhance the paper's clarity and understanding.

**Additional Comments On Reviewer Discussion:**

During the discussion, the authors addressed the reviewers' concerns with additional experiments and clarifications. They acknowledged PRNS's concerns about handling complex, text-rich images and explained that increasing the parameter α shifts the model toward being a reward model, which can reduce language generation capabilities. To KRLQ, they clarified TLDR's role as a reward model in RLHF contexts and provided empirical evidence supporting the questioned equation. They addressed f5mm's concerns by conducting experiments with stronger models, including more tasks and models in their evaluation, and agreeing to discuss limitations regarding state-of-the-art models. For UgXE, they highlighted their novel contributions and added missing references. Considering these responses and the effective resolution of the reviewers' concerns, this paper contributes valuable advancements to the field.

---

### Decision · Program_Chairs · 2025-01-22

Accept (Poster)